# Enhanced CAR-T activity against established tumors by polarizing human T cells to secrete interleukin-9

Lintao Liu[1,3], Enguang Bi[1,3 ✉], Xingzhe Ma[1,3], Wei Xiong[1,3], Jianfei Qian[1], Lingqun Ye[1], Pan Su[1], Qiang Wang[1], Liuling Xiao[1], Maojie Yang[1], Yong Lu[2] & Qing Yi [1 ✉]

CAR-T cell therapy is effective for hematologic malignancies. However, considerable numbers of patients relapse after the treatment, partially due to poor expansion and limited persistence of CAR-T cells in vivo. Here, we demonstrate that human CAR-T cells polarized and expanded under a Th9-culture condition (T9 CAR-T) have an enhanced antitumor activity against established tumors. Compared to IL2-polarized (T1) cells, T9 CAR-T cells secrete IL9 but little IFN-γ, express central memory phenotype and lower levels of exhaustion markers, and display robust proliferative capacity. Consequently, T9 CAR-T cells mediate a greater antitumor activity than T1 CAR-T cells against established hematologic and solid tumors in vivo. After transfer, T9 CAR-T cells migrate effectively to tumors, differentiate to IFN-γ and granzyme-B secreting effector memory T cells but remain as long-lived and hyperproliferative T cells. Our findings are important for the improvement of CAR-T cell-based immunotherapy for human cancers.

[1] Center for Translational Research in Hematologic Malignancies, Houston Methodist Cancer Center/Houston Methodist Research Institute, Houston Methodist, Houston, TX 77030, USA. [2] Comprehensive Cancer Center, Wake Forest Baptist Health, and Department of Microbiology & Immunology, Wake Forest School of Medicine, Winston-Salem, NC, USA. [3]These authors contributed equally: Lintao Liu, Enguang Bi, Xingzhe Ma, Wei Xiong.
✉email: bienguang1980@smu.edu.cn; qyi@houstonmethodist.org

Immunotherapy with chimeric antigen receptor (CAR) T cells is a promising strategy to improve therapeutic outcomes in hematological malignancies[1,2]. In the past decade, great successes have been achieved in research and clinical trials of CAR-T cell therapy[3–6]. Although various CAR-T cells targeting different antigens have been developed, CD19-targeted or BCMA-targeted CAR-T cell therapy has proven to be most effective against relapsed/refractory diffuse large B-cell lymphoma, acute lymphoblastic leukemia, or multiple myeloma, respectively[7,8]. However, considerable numbers of patients relapse after CAR-T cell treatment, even for patients who achieved complete remission at early stages of treatment[9]. The median overall survival is 2 years for patients with relapsed or refractory large B-cell lymphoma[10]. For patients with acute lymphoblastic leukemia, the median overall survival is only about 19 months[11]. Six-month overall survival rate for patients with acute lymphoblastic leukemia is <80%[12].

The causes for treatment failure include, but are not limited to, antigen loss[13,14] and poor persistence[12] of infused CAR-T cells. There are two major reasons that account for the poor persistence of CAR-T cells, programmed cell death[15] and exhaustion[12,16]. Exhaustion of T cells is a more complex phenomenon than programmed cell death and can be affected by a variety of factors, such as the starting state of T cells[17], patient age[18], and tumor microenvironment[19].

Recently, we[20–23] and others[24,25] have shown that antigen-specific IL9-secreting (CD4+ Th9 or CD8+ Tc9) T cells are distinct subsets with stronger antitumor efficacy in murine tumor models compared to Th1, Th17, or Tc1/CTLs, respectively. These subsets express different cytokine profiles and low cytolytic proteins and exhaustion markers compared to other subsets of T cells[20–22]. Therefore, we hypothesized that human CAR-T cells polarized ex vivo under a Th9-cultured condition may also be more effective than CAR-T cells polarized under the classical Th1/Tc1-cultured condition to eradicate established human tumors in vivo.

In this work, we provide a method for CAR-T cell polarization and expansion. We polarize and expand CD19 or GPC3 CAR-T cells under Th9 condition. The results show that IL9-secreting CAR-T cells exhibit distinguished features compared to IFN-γ-secreting CAR-T cells, including less differentiation state and hyperproliferative capacity both in vitro and in vivo. Consequently, these CAR-T cells exert a greater antitumor efficacy against CD19-expressing human acute lymphoblastic leukemia or GPC3-expressing liver carcinoma in vivo.

## Results

### Human IL9-secreting CAR-T cells display distinct cytokine expression profiles.

We previously demonstrated that murine IL9-producing T cells displayed strong antitumor efficacy and maintained long persistence in vivo[20–22]. To assess whether human CAR-T cells possess such features in culture supplemented with IL4 and TGF-β (Th9 polarization condition), purified human CD3+ T cells were stimulated with anti-CD3/CD28 beads, polarized under Th9-culture condition (T9), and then transduced with a second-generation CAR that contains a single chain variable fragment (scFv) recognizing human CD19, hinge plus transmembrane CD8, 4-1BB, and CD3ζ intracellular signaling domains. T cells polarized with IL2 were used as T1 CAR-T cell controls[26]. Flow cytometry showed that T cells polarized under T9 exhibited higher transduction efficacy compared to T cells under Th1-polarization condition (Supplementary Fig. 1a–c).

First, we defined the signature cytokines of T9 CAR-T cells. After 16-day expansion in vitro, mRNA was extracted from sorted CD4+ or CD8+ CAR-T cells. Real-time PCR results showed that T9 CAR-T cells expressed high levels of IL9 and low levels of IFN-γ, whereas T1 CAR-T cells expressed low levels of IL9 and high levels of IFN-γ (Fig. 1a, b), which were confirmed by flow cytometry analysis (Fig. 1c, d, Supplementary Fig. 2). To assess cytokine release from CAR-T cells, sorted Th9 or Tc9 CAR-T cells were stimulated by antigen-expressing tumor cells (CD19-expressing K562 cells; wild-type K562 cells served as a negative control). We observed that exposure to tumor cells significantly enhanced the production of IL9 by Th9 and Tc9 but not Th1 or Tc1 CAR-T cells (Fig. 1e). Interestingly, tumor exposure also promoted IFN-γ production by Th9 and Tc9 CAR-T cells, although less than that of Th1 or Tc1 CAR-T cells. In addition, Th9 CAR-T cells secreted lower amounts of IL2 and TNF-α than Th1 CAR-T cells, but Tc9 CAR-T cells secreted significantly higher levels of IL2 and TNF-α than Tc1 CAR-T cells, especially after coculture with tumor cells. Furthermore, T9 CAR-T cells secreted much fewer IL4 and IL10 than T1 CAR-T cells (Fig. 1e).

To further characterize their cytokine and cytokine receptor profiles, we analyzed the transcriptome of human Th9 or Th1 CAR-T cells by RNASeq analysis. Cluster analysis showed that Th9 CAR-T cells had a distinct gene signature from Th1 CAR-T cells. Among the differentially expressed genes, those with log2 value of fold change over 2 or less than −2 were filtered. Volcano plot showed a differential expression of nearly 1300 transcripts, including 641 upregulated genes and 608 downregulated genes in Th9 CAR-T cells compared to Th1 CAR-T cells (Supplementary Fig. 3a, b). In addition to IL9 and IFN-γ, Th9 CAR-T cells expressed completely different profiles of cytokine, chemokine, and their receptors compared to Th1 CAR-T cells (Fig. 1f), suggesting that functionality of Th9 CAR-T cells is significantly different from that of Th1 CAR-T cells.

We further examined whether the difference in cytokine secretion would affect the killing ability of T9 CAR-T cells. Cytotoxicity assay was performed by coculturing CAR-T cells with tumor cells for 24 h at different effector:target cell ratios. T9 CAR-T cells exhibited similarly strong cytolytic activity against target tumor cells (CD19-expressing K562 and CD19+ NALM6 cells) as T1 CAR-T cells did (Fig. 1g). No killing of control K562 cells was observed.

### Human T9 CAR-T cells manifest robust proliferative capacity.

During in vitro expansion period, we observed that both CD4+ and CD8+ T cells cultured under the Th9-polarization condition increased in cell size, followed by a decrease in cell size at a slower pace than cells polarized under the Th1 condition (Supplementary Fig. 4), and significantly increased proliferative capacity over Th1 or Tc1 cells (Fig. 2a, b). Additionally, fewer fractions of Th9 or Tc9 CAR-T cells were stained with the apoptosis marker annexin V than in Th1 or Tc1 CAR-T cells (Fig. 2c).

To understand the mechanisms contributing to the decreased apoptosis and hyperproliferative capacity of T9 CAR-T cells, genes associated with apoptosis, cell cycle, and proliferation were analyzed by gene-set enrichment analysis (GSEA). The results showed that while Th1 cells had higher transcript levels of genes associated with apoptosis (Fig. 2d), Th9 cells had higher expression of genes encoding G2/M transition (Fig. 2e), DNA replication (Fig. 2f), and cell cycle (Fig. 2g). Heatmap analysis showed that the expression of minichromosome maintenance (MCM) transcripts, including MCM2, MCM4, and MCM6, was much higher in Th9 cells (Fig. 2h), suggesting that the MCM2-7 complexes, which serve as replicative helicases[27], in Th9 cells are functionally distinct from Th1 cells. Moreover, Th9 CAR-T cells were also enriched in CDK2 (Fig. 2h), which participates in cell cycle regulation[28].

We proceeded to assess the proliferative capacity of T9 or T1 CAR-T cells after exposure to CD19-expressing tumor cells (NALM6 cells). Anti-CD3/CD28 beads were used as control. T9 and T1 CAR-T cells were cocultured with indicated tumor cells at a ratio of 1:1. Four days later, gated GFP$^+$CD3$^+$ CD4$^+$ (Th9) or

CD8$^+$ (Tc9) CAR-T cells were enumerated using CountBright™ absolute counting beads. Both Th9 and Tc9 CAR-T cells showed robust proliferative capacity in culture with anti-CD3/CD28 beads or NALM6 cells, which was at least 5-fold higher than their counterparts, Th1 and Tc1 CAR-T cells (Fig. 2i).

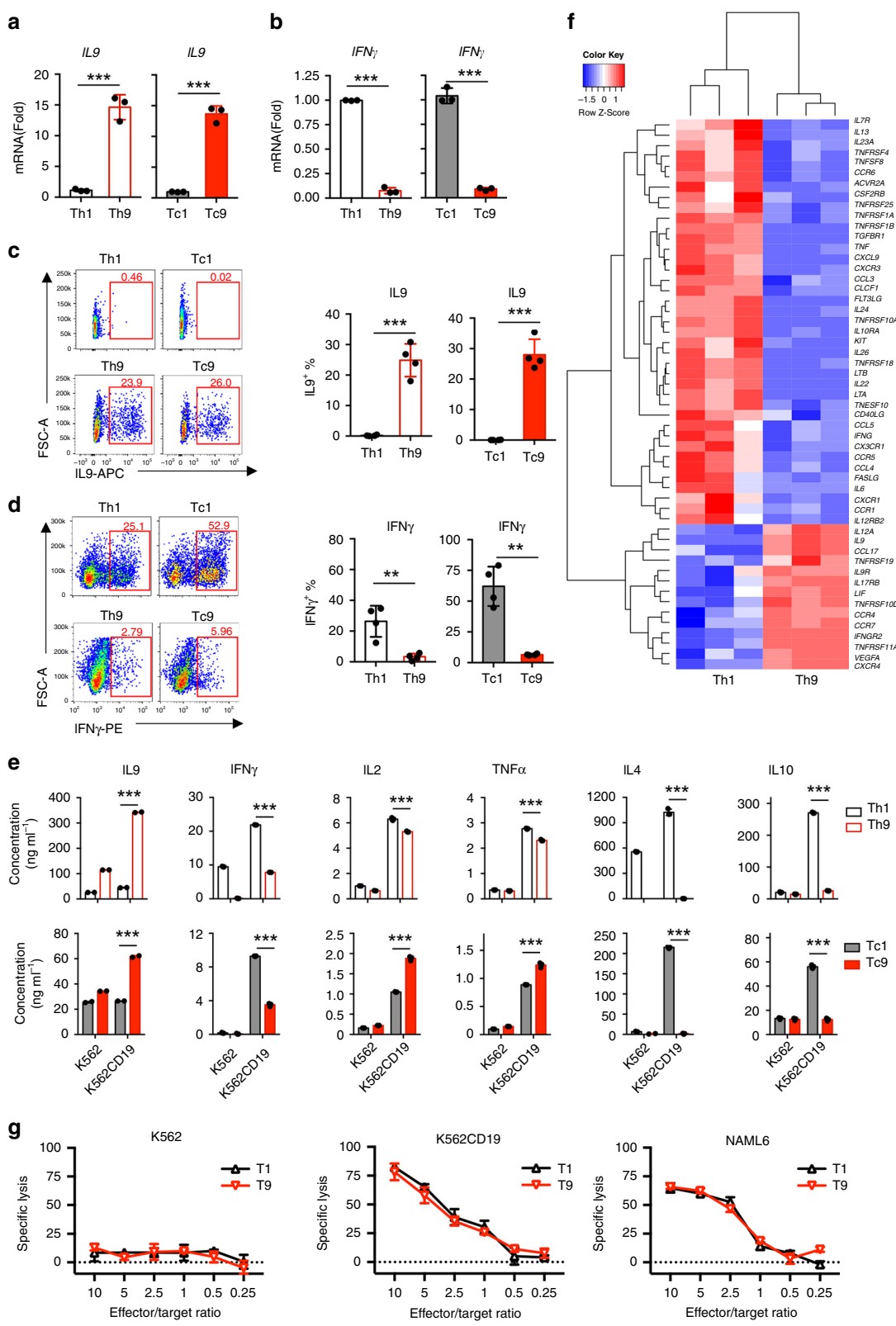

**Fig. 1 Characterization of T9 CAR-T cell cytokine expression profiles.** CAR-T cells were harvested and analyzed at day 16 during in vitro expansion. Real-time PCR analysis of relative mRNA expression of **a** IL9 ($n = 4$ donors) or **b** ($n = 4$ donors) IFN-γ in sorted CD4$^+$ and CD8$^+$ CAR-T cells. **c** Representative flow plots (left panels) and summarized data (right panels; $n = 4$ donors) showing IL9 expression in CAR-T cells. Cells were pre-gated for GFP$^+$CD3$^+$CD8$^+$ or GFP$^+$CD3$^+$CD4$^+$ T cells. **d** Representative flow plots (left panels) and summarized data (right panels; $n = 4$ donors) showing IFN-γ expression in CAR-T cells. Cells were pre-gated for GFP$^+$CD3$^+$CD8$^+$ or GFP$^+$CD3$^+$CD4$^+$ T cells. **e** Concentrations of secreted cytokines by sorted Th1 or Th9 (upper panels), or Tc1 or Tc9 (lower panels) CAR-T cells in response to K562 or CD19-expressing K562 (K562-CD19) tumor cells. Production of indicated cytokines was determined by BD Cytometric Bead Assay kit ($n = 3$ donors). **f** Heatmap illustrating the relative expression of cytokines and cytokine receptors by in vitro polarized Th9 or Th1 CAR-T cells. RNA of purified CD4$^+$ GFP$^+$ CAR-T cells was extracted for RNASeq. **g** Percentage of specific lysis of CAR-T cells against target tumor cells at indicated E:T ratios determined by luciferase assay at 24 h culture. Experiments were performed with three biological replicates and data shown are representative of three independent experiments. Data are presented as mean ± SD (*$P < 0.05$, **$P < 0.01$, and ***$P < 0.001$, two-sided Student's $t$-test). Source data are provided as a Source Data file.

**Human T9 CAR-T cells are less differentiated and exhausted T cells.** On the basis of differences in the cytokine profile and proliferative capacity, we hypothesized that T9 CAR-T cells would have a different differentiation state than T1 CAR-T cells. To accurately characterize the differentiation state of Th9 CAR-T cells in an unbiased manner, we again performed GSEA of the RNASeq data. Interestingly, genes involved in central memory (T$_{cm}$) cells were significantly enriched in Th9 CAR-T cells (Fig. 3a), whereas genes for effector memory (T$_{em}$) cells were significantly enriched in Th1 CAR-T cells (Fig. 3b). These results were confirmed by flow cytometry examining the expression of CCR7 and CD45RO on both CD4$^+$ and CD8$^+$ T cells after 16-day in vitro expansion. T9 CAR-T cells maintained a higher proportion of CCR7$^+$CD45RO$^+$ T$_{cm}$ cells, while T1 CAR-T cells had larger populations of CCR7$^-$CD45RO$^+$ T$_{em}$ cells compared to their counterparts (Fig. 3c, d).

Different from T$_{em}$ cells, T$_{cm}$ cells have unique granzyme production and transcriptional profiles[29]. Transcriptional expression of lytic granule proteins was compared in CD4$^+$ CAR-T cells. Human Th9 CAR-T cells expressed fewer cytotoxic molecules such as *PRF1*, *FAS*, and granzymes (*GZMA*, *GZMB*, *GZMK*, *GZMM*) (Fig. 3e, f), which suggests that T9 CAR-T cells represent a less terminally differentiated subset[30]. Protein level of granzyme B (GZMB) was assessed by flow cytometry, which showed that Th9 and Tc9 CAR-T cells had a significantly lower expression of GZMB compared to Th1 and Tc1 CAR-T cells (Supplementary Fig. 5). RNASeq analysis showed that Th9 CAR-T cells had reduced expression of genes regulating effector differentiation, such as PU.1 (*SPI1*), eomesodermin (*EOMES*), T-box 21 (*TBX21*), and *IRF1* compared to Th1 CAR-T cells (Fig. 3g, h). Tc9 CAR-T cells displayed similar gene expression profiles as Th9 CAR-T cells (Fig. 3f–h). Both Th9 and Tc9 cells expressed higher levels of *IRF4* compared to their counterparts (Fig. 3g, h).

We also analyzed exhaustion-related genes of the T cells using our RNASeq data and found that JUN, which was reported to induce exhaustion resistance in CAR-T cells[31], was highly expressed in Th9 but not Th1 CAR-T cells (Supplementary Fig. 6a). To confirm this result, Th9 CAR-T cells expanded at day 14 were subjected to western blot analysis, which showed that Th9 cells exhibited not only higher c-jun expression but also c-jun phosphorylation at Ser73 (Supplementary Fig. 6b).

Next, to determine which cytokine plays a major role in shaping the properties of T9 CAR-T cells, we polarized CD3$^+$ CAR-T cells under four different conditions (none, IL4, TGFβ, IL4 + TGFβ). IL2 was added to all of the culture to support T cell survival and growth. Results showed that TGFβ alone induced the expression and phosphorylation of c-jun (Supplementary Fig. 6c) and expression of CCR7 (Supplementary Fig. 6d). However, TGFβ alone significantly suppressed cell growth, which could be rescued by addition of IL4 (Supplementary Fig. 6e). In addition, IL4 reduced the percentage of PD1$^+$TIM3$^+$ population (Supplementary Fig. 6f). Therefore, the property of T9 CAR-T cells,

including IL9 expression, is the result of a combined effect of TGFβ and IL4 (Supplementary Fig. 6g).

It has been demonstrated that T$_{cm}$ cells have significantly enhanced respiratory capacity[32]. We therefore examined oxygen consumption of CAR-T cells. The results showed a robust increase in oxygen consumption rate (OCR), maximal respiratory capacity and basal OCR in Th9 compared to Th1 CAR-T cells (Supplementary Fig. 7a–c). Taken together, T9 CAR-T cells express low levels of T-cell exhaustion markers (PD-1 and TBX-21) and more CCR7 that is associated with T$_{cm}$ cells, indicating that T9 CAR-T cells represent less terminally differentiated T cells.

**Human T9 CAR-T cells display strong antitumor activity in vivo.** Given the observed phenotype and proliferative capacity of T9 CAR-T cells, we next asked whether T9 CAR-T cells have an enhanced antitumor ability in vivo. NALM-6 tumor cells were intravenously injected into NSG mice and 7 days later when tumor burden developed, mice received $4 \times 10^6$ CD3$^+$ T9 or T1 CAR-T cells via the tail vein (Fig. 4a). T9 CAR-T cells were more effective in significantly reducing tumor burden (Fig. 4b, c) and prolonging the survival of tumor-bearing mice compared to T1 CAR-T cells (Fig. 4d).

To elucidate the mechanisms underlying the significantly greater in vivo antitumor capacity of T9 CAR-T cells, we first assessed the persistence of these cells in vivo by analyzing blood and spleen (the primary tumor-infiltrating organ) for the presence and number of CAR-T cells over the time course of tumor growth. Flow cytometry analysis revealed that in both peripheral blood (from days 4 to 24) (Fig. 4e, f) and spleen (from days 4 to 16) (Fig. 4g), the absolute numbers (Fig. 4f, g) of CD3$^+$ (left panels) or CD4$^+$ (middle panels) and CD8$^+$ (right panels) CAR-T cells were significantly higher in T9 CAR-T-treated mice than those treated with T1 CAR-T cells. In addition to spleen, NALM6 cells also metastasize to the bone marrow[33]. Therefore, we examined CAR-T infiltration into bone marrow and observed that more T9 (compared to T1) CAR-T cells were found in the bone marrow of tumor-bearing mice at day 8 after transfer (Fig. 4h). These results demonstrate that the better persistence and tumor infiltration abilities of T9 CAR-T cells contribute to the greater antitumor efficacy in vivo.

To determine which type of CAR-T cells contribute more to antitumor effect in vivo, we isolated human CD4$^+$ and CD8$^+$ T cells, polarized them in vitro under Th9- or Th1-culture condition, and injected them into tumor-bearing mice (Supplementary Fig. 8a). Our results show that Tc9 CAR-T cells exhibited the strongest antitumor efficacy compared to Th9, Th1, or Tc1 CAR-T cells, and Th9 CAR-T cells had similar antitumor ability as Tc1 cells but better effect than Th1 cells in vivo (Supplementary Fig. 8b–d). In addition, Th9 or Tc9 CAR-T cells showed stronger proliferation ability than Th1 and Tc1 CAR-T cells (Supplementary Fig. 8e).

**Human T9 CAR-T cells switch into Th1-like effector cells in vivo**. Next, we analyzed the expression of T-cell effector molecules in CAR-T cells on days 8 and 12 post transfer. Compared to in vitro polarized cells that expressed high levels of IL9 and low levels of IFN-γ, IL9 expression in CD4$^+$ Th9 CAR-T cells from blood and spleen was barely detected on day 8 and absent on day 12, while IL9 expression in CD8$^+$ Tc9 CAR-T cells was

undetected on both days (Fig. 5a). Remarkably, both Th9 and Tc9 cells isolated from blood and spleen started to produce similar amounts of IFN-γ on day 8 and maintained higher production of IFN-γ than Th1 and Tc1 CAR-T cells did on day 12 (Fig. 5a, b). In addition, the frequency of GrzB-positive Th9 and Tc9 CAR-T cells from blood and spleen was also dramatically increased after transfer (Fig. 5c, d). Although IL2 production by in vitro-

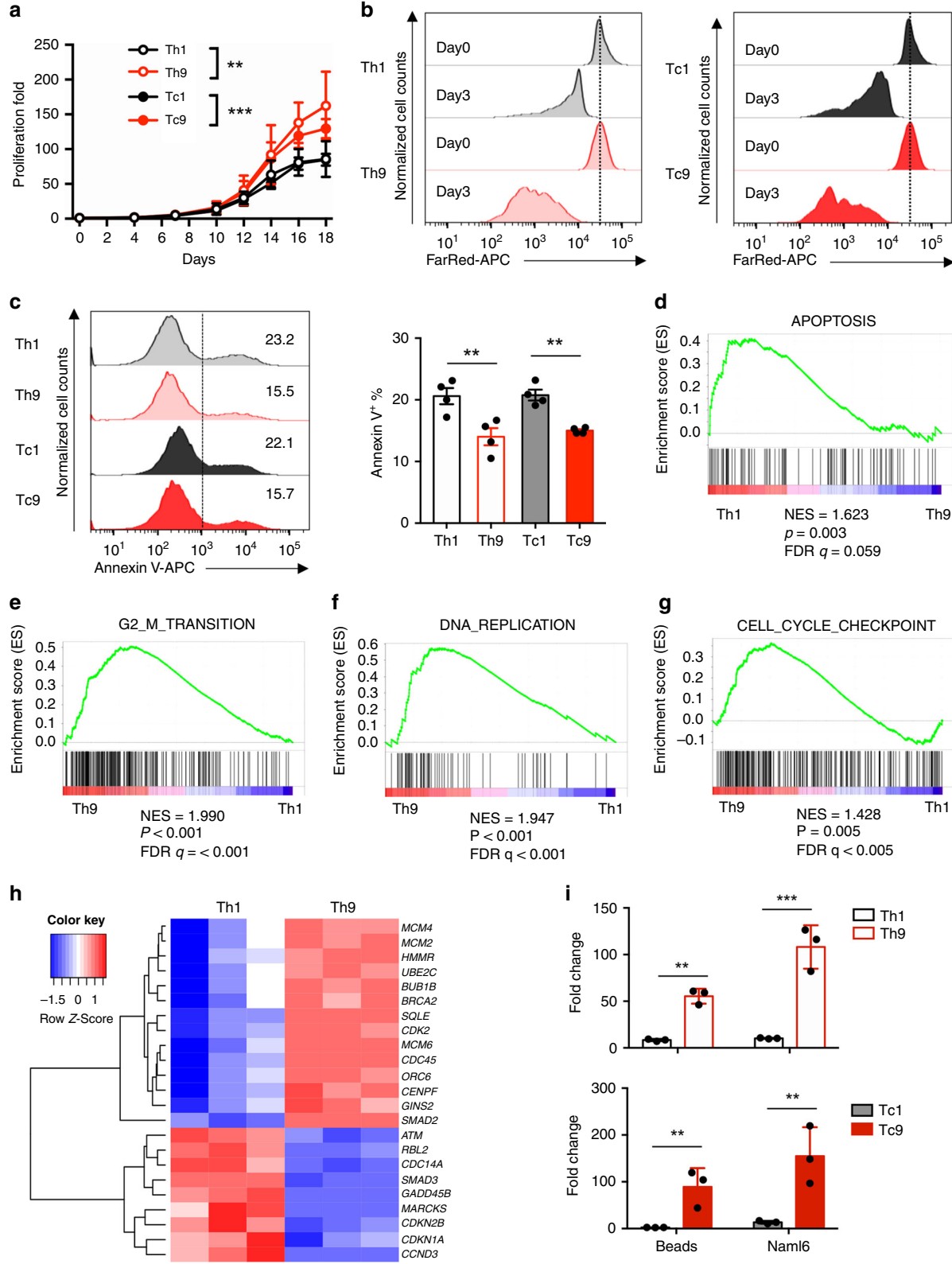

**Fig. 2 T9 CAR-T cells possess a unique hyperproliferative advantage over T1 CAR-T cells. a** Number of viable CAR-T cells determined by trypan blue exclusion. Fold expansion of CAR-T cells was calculated from day 0 to day 18. Bars indicate the mean ± SD ($n = 4$ donors, two-way ANOVA). **b** FarRed dilution assay performed at day 14 with T1 or T9 CAR-T cells after first stimulation with anti-CD3/CD28 beads, and FarRed staining intensity was analyzed 72 h after staining. Cells were pre-gated for GFP$^+$CD3$^+$CD4$^+$ (Th1 or Th9) (left panels), or GFP$^+$CD3$^+$CD8$^+$ (Tc1 or Tc9) (right panels) T cells. Data are representative of three independent experiments. **c** Representative flow plots (left panels) and summarized results (right panels; $n = 4$ donors) of annexin V$^+$ apoptotic CAR-T cells at day 16 of expansion. GSEA results of **d** KEGG apoptosis gene, **e** G2M checkpoint gene, **f** DNA replication gene, and **g** cell cycle gene signatures. NES normalized enrichment score, FDR false discovery rate. **h** Heatmap illustrating the relative expression of genes involved in G2M checkpoint, DNA replication and cell cycle regulation. **i** Number or fold change of Th1 or Th9 (upper panels), or Tc1 or Tc9 (lower panels) CAR-T cells in culture with beads or NALM6 tumor cells for 4 days. GFP$^+$CD3$^+$CD4$^+$ cells and GFP$^+$CD3$^+$CD8$^+$ T cells were counted with CountBright$^{TM}$ Counting Beads. Fold change was calculated relative to the number of cells without the stimuli ($n = 3$ donors). Data are presented as mean ± SD (*$P < 0.05$, **$P < 0.01$, and ***$P < 0.001$, two-sided Student's $t$-test). Source data are provided as a Source Data file.

polarized Th9 CAR-T cells was significantly lower than Th1 CAR-T cells, Th9 CAR-T cells gradually increased IL2 production after transfer and maintained its production levels equal to those produced by Th1 CAR-T cells (Fig. 5e, f). These results indicate that T9 CAR-T cells acquire an effector T-cell function after transfer.

To determine the role of IL9 or IFN-γ on the efficacy of T9 CAR-T cells, we treated NALM6-bearing mice with cytokine neutralizing antibodies followed by injection of T9 CAR-T cells (Supplementary Fig. 9a). Anti-IL9 or anti-IFN-γ antibodies did not significantly affect T9 CAR-T-mediated antitumor efficacy in vivo (Supplementary Fig. 9b–d). We also assessed the persistence of T9 CAR-T cells in treated mice and revealed that the antibodies did not affect the numbers of CAR-T cell numbers in blood (Supplementary Fig. 9e).

**Human T9 CAR-T cells evolve into effector but remain less exhausted T cells in vivo.** After determining that T9 CAR-T cells evolved into Th1-like effector T cells, we explored whether they also displayed a maturation phenotype similar to T1 CAR-T cells. We analyzed CAR-T cell phenotype by examining the expression level of CCR7 and CD45RO from days 4 to 16 after transfer. The percentages of T$_{cm}$ (CCR7$^+$CD45RO$^+$) subsets in Th9 and Tc9 CAR-T cells decreased dramatically to levels similar to Th1 and Tc1 CAR-T cells on day 8 (Fig. 6a, b), and the percentages of T$_{em}$ (CCR7$^-$CD45RO$^+$) subsets in Th9 and Tc9 CAR-T cells reached peaks on day 8 and decreased to the low levels similar to Th1 and Tc1 CAR-T cells on day 16 (Fig. 6a, b). However, the absolute numbers of these cells were significantly higher in T9 CAR-T cells compared to T1 CAR-T cells (Fig. 6c). T$_{emra}$ (terminally differentiated effector memory; CCR7$^-$CD45RO$^-$) subset also increased remarkably after transfer in T1 and T9 CAR-T cells, but its frequency in T9 CAR-T cells remained lower than that in T1 CAR-T cells (Fig. 6a, b). The absolute number of T$_{emra}$ subset in T9 CAR-T cell-treated mice peaked at day 8 and then began to decrease (Fig. 6c). Mitochondria staining by MitoTracker revealed that mitochondria mass of T9 CAR-T cells was larger than that of T1 CAR-T cells (Supplementary Fig. 10a, b).

Given the differences in subsets, we speculated that T9 CAR-T cells would have downregulated expression of co-inhibitory receptors, such as PD1, TIM3, and LAG3. As expected, both Th9 and Tc9 CAR-T cells expressed significantly lower levels of PD1 compared to Th1 or Tc1 CAR-T cells, and Tc9 CAR-T cells expressed lower levels of LAG3 than Tc1 CAR-T cells (Fig. 6d, e).

Next, we examined the percentages of apoptotic CAR-T cells by annexin V staining and showed that T9 CAR-T cell population had significantly fewer annexin V$^+$ cells than those of T1 CAR-T cells (Fig. 6f). Taken together, these data indicate that T9 CAR-T cells exhibit a higher proliferative and less exhausted phenotype, resulting in better antitumor ability compared to T1 CAR-T cells in vivo.

**Human T9 CAR-T cells show enhanced capacity to suppress solid tumor growth.** Finally, we investigated the efficacy of T9 CAR-T cells to treat solid tumors. We established a solid tumor model by subcutaneous injection of NALM6 cells into NSG mice. After 14 days, tumor-bearing mice were treated with CAR-T cells (Fig. 7a). Compared to T1 CAR-T cells, T9 CAR-T cells significantly inhibited tumor growth (Fig. 7b) and improved the overall survival of mice (Fig. 7c). The improved therapeutic efficacy of T9 CAR-T cells was associated with an increased frequency of CAR-T cells in peripheral blood, which was approximately 5-fold more than in mice treated with T1 CAR-T cells (Fig. 7d, e). More importantly, significantly more tumor-infiltrating GFP$^+$CD3$^+$ CAR-T cells were detected in T9 CAR-T-treated mice than in mice treated with T1 CAR-T cells (Fig. 7f, g), indicating that more T9 CAR-T cells infiltrated into tumor sites.

To confirm the results, we constructed GPC3-41BB-Z CAR in retroviral vector and generated GPC3-targeted CAR-T cells. Th9-polarizing condition also significantly increased GPC3 CAR expression in T cells (Supplementary Fig. 11a). Flow cytometry showed that GPC3 Th9 and Tc9 CAR-T cells exhibited significantly enhanced IL9 (Supplementary Fig. 11b) and reduced IFN-γ (Supplementary Fig. 11c) expressions. GPC3 T9 CAR-T cells expressed high level of CCR7, indicating higher percentage of T$_{cm}$ (CCR7$^+$CD45RO$^+$) cells compared to GPC3 T1 CAR-T cells (Supplementary Fig. 11d).

To determine antitumor activity of T1 and T9 GPC3 CAR-T cells, we first performed in vitro cytotoxicity assay. Both T1 and T9 GPC3 CAR-T cells effectively killed GPC3-expressing HepG2 cells (Supplementary Fig. 11e). Next, HepG2 cells were subcutaneously injected into NSG mice, and tumor-bearing mice were treated with CAR-T cells 14 days after tumor inoculation (Fig. 7h). Both T1 and T9 GPC3 CAR-T cells suppressed HepG2 growth, however, T9 GPC3 CAR-T cells exhibited much stronger antitumor activity than T1 GPC3 CAR-T cells (Fig. 7I, j). At days 8 and 12, we detected CAR-T cells in peripheral blood of treated mice. Higher frequency of T9 than T1 GPC3 CAR-T cells were detected at both days (Fig. 7k, l). Importantly, more T9 than T1 GPC3 CAR-T cells were also detected in tumors (Fig. 7m, n).

Altogether, these data suggest that tumor-specific T9 CAR-T cells may be able to mediate effective antitumor immunity against solid tumors due to their robust proliferative and better tumor infiltration capacities.

**Discussion**
CAR-T cell therapy has been shown to be an effective cancer immunotherapy for different types of human cancer. However, long-term remission is still uncommon and most patients relapse[9]. Therefore, there is an urgent need to improve the therapeutic efficacy of CAR-T cells. In our study, we evaluated the antitumor efficacy of human IL9-secreting (T9) CAR-T cells in comparison with conventional IFN-γ-secreting (T1) CAR-T cells

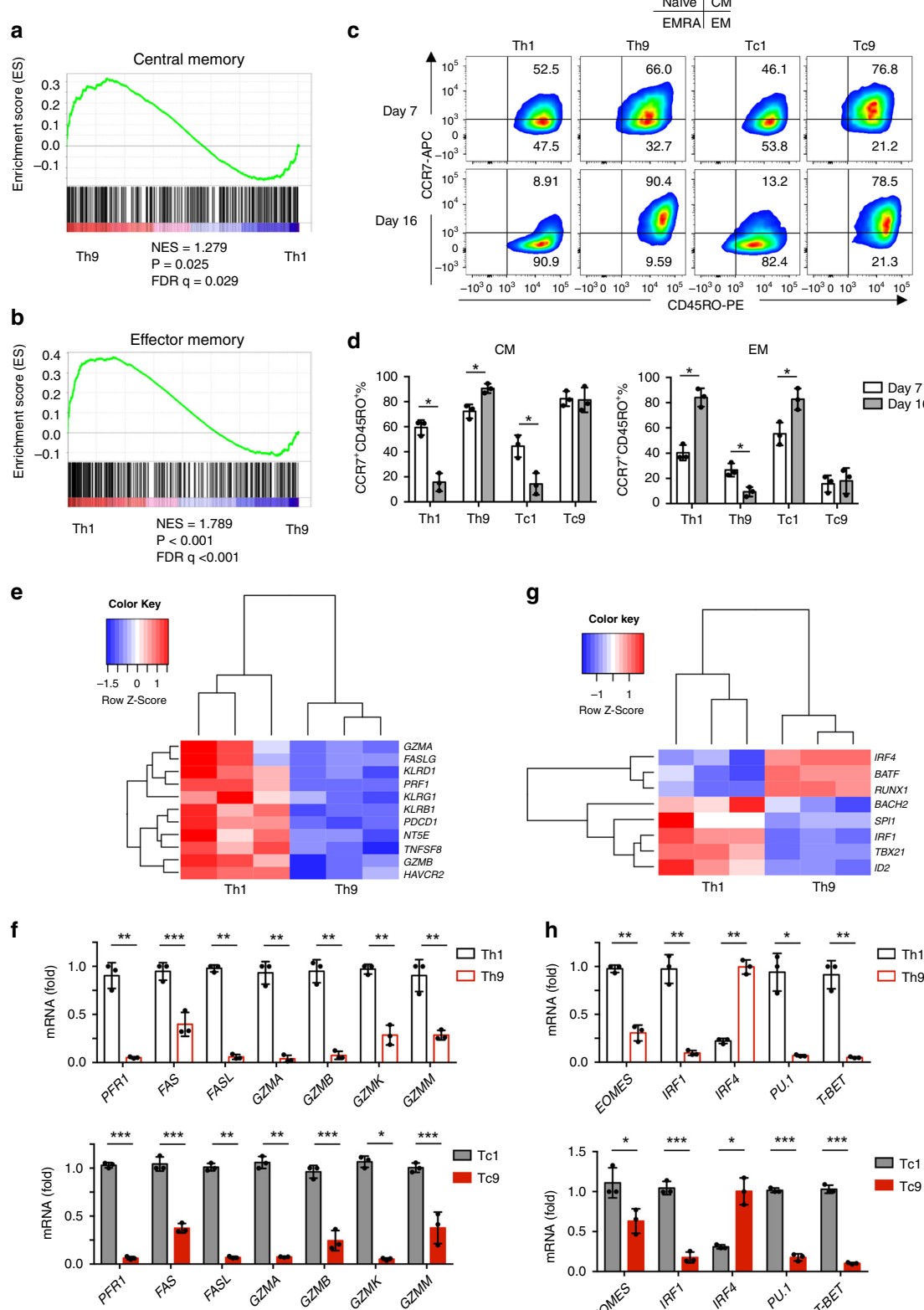

**Fig. 3 In vitro generated T9 CAR-T cells exhibit T_cm-like and less exhausted phenotypes.** GSEA results showing **a** central memory or **b** effector memory gene signatures of polarized Th9 or Th1 cells. **c** Representative flow plots and **d** pie charts ($n = 3$ donors) showing CCR7 or CD45RO expression on CAR-T cells at days 7 and 16. Cells were pre-gated for GFP+CD3+CD4+ or GFP+CD3+CD8+ T cells. **e** Heatmap showing the expression of exhaustion and cytolytic genes in Th9 or Th1 CAR-T cells assessed by RNASeq. **f** Expression levels (relative to GAPDH) of exhaustion and cytolytic genes in Th1 or Th9 (upper panels), and Tc1 or Tc9 (lower panels) CAR-T cells measured by qPCR. **g** Heatmap illustrating the expression of genes of transcription factors in Th9 or Th1 CAR-T cells. **h** Expression levels (relative to GAPDH) of transcription factor genes in Th1 or Th9 (upper panels), and Tc1 or Tc9 (lower panels) CAR-T cells measured by qPCR, **f** and **h** were performed with three biological replicates and data shown are representative of two independent experiments. Data are presented as mean ± SD. *$P < 0.05$, **$P < 0.01$, and ***$P < 0.001$, two-sided Student's $t$-test. Source data are provided as a Source Data file.

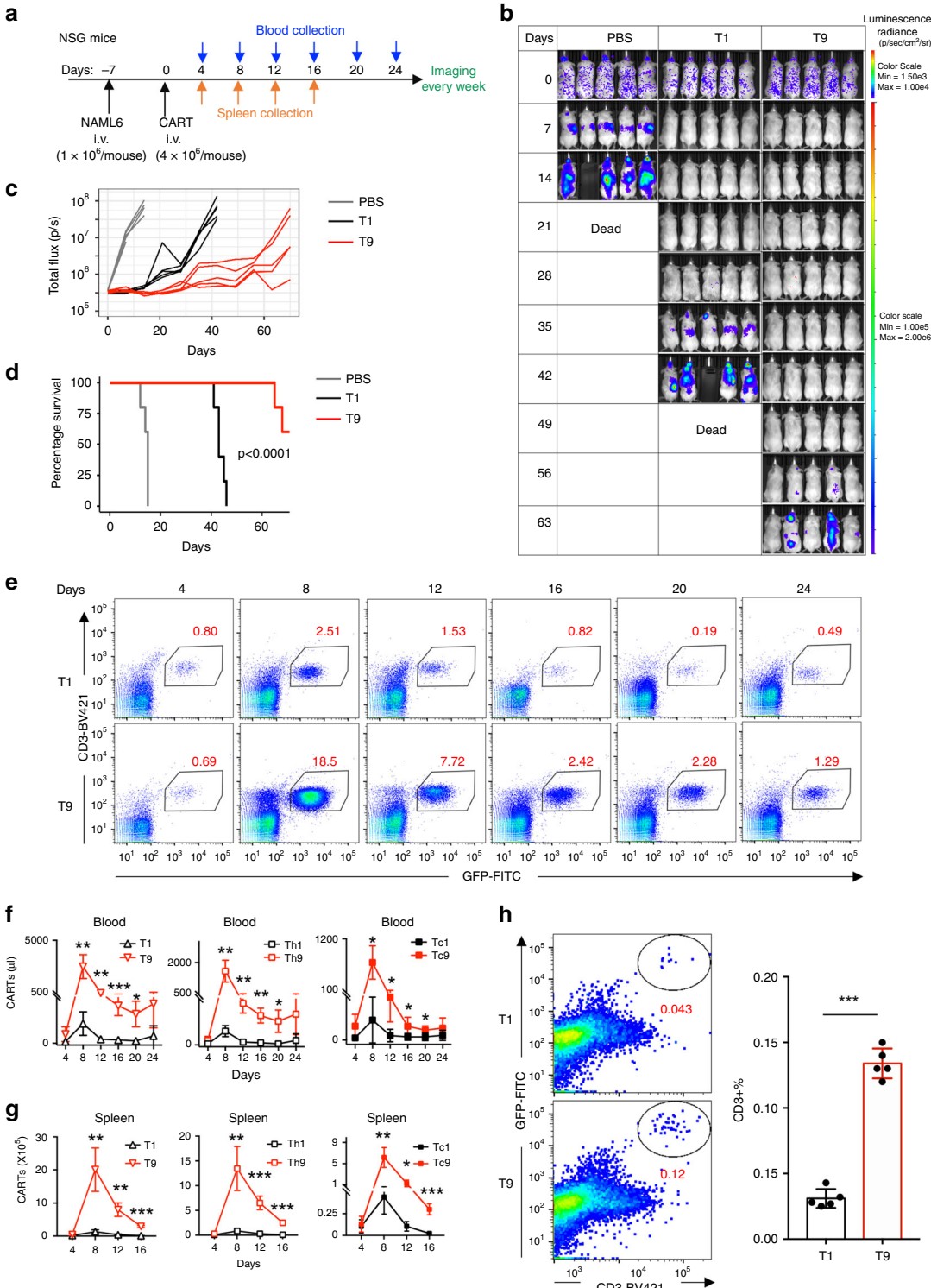

**Fig. 4 T9 CAR-T cells exhibit enhanced antitumor activity against established tumor in vivo. a** NSG mice were intravenously injected with $1 \times 10^6$ NALM6 cells. Seven days later, mice were randomly assigned to three groups and were infused intravenously with $4 \times 10^6$ CD3$^+$ CAR-T cells. **b** Tumor burden measured by bioluminescence at indicated days since CAR-T cell infusion. **c** Tumor burden (total flux) quantified by photons/s in mice treated with PBS or CAR-T cells at indicated days since CAR-T cell infusion. **d** Kaplan–Meyer plot showing mouse survival. Exact *p* values from log-rank test are shown for T1 versus T9 CAR-T cell-treated mice, and data shown are representative of two independent experiments. **e** Representative flow plots showing the frequency of CD3$^+$ CAR-T cells in blood of treated mice at day 8 after CAR-T infusion. Total number of CD3$^+$ CAR-T cells in **f** blood ($n = 5$ mice) or **g** spleen ($n = 4$ mice at day 4, $n = 5$ mice at other days) of treated mice at different days since CAR-T infusion. **h** Percentage of bone marrow-infiltrating CAR-T cells identified by CD3$^+$GFP$^+$ T cells at day 8 after transfer. Representative plots (left panels) and summarized results ($n = 5$ mice; right panels) are shown. Data are presented as mean ± SD. *$P < 0.05$, **$P < 0.01$, and ***$P < 0.001$, two-sided Student's *t*-test. Source data are provided as a Source Data file.

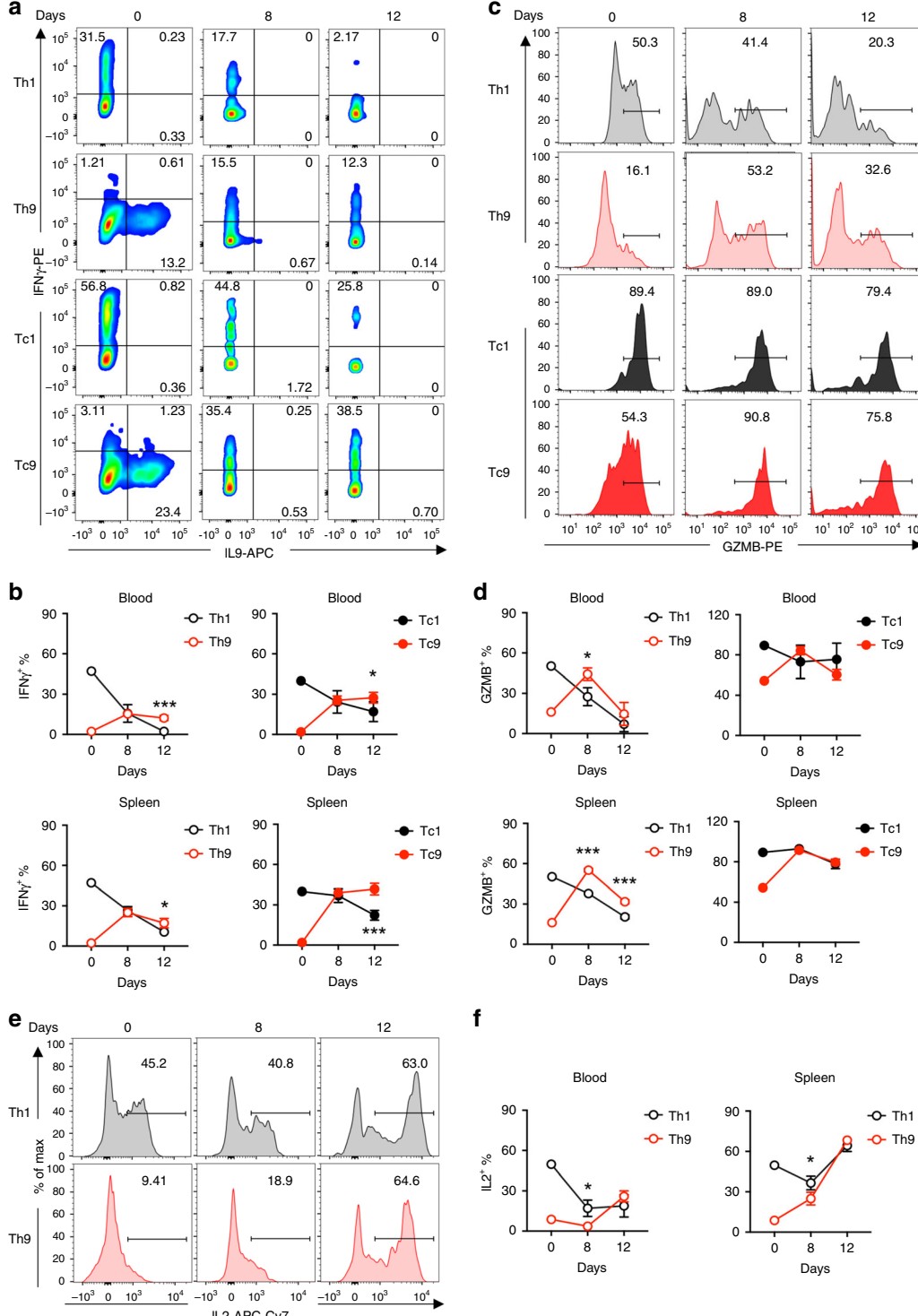

**Fig. 5 T9 CAR-T cells acquire effector function in vivo after transfer. a** Representative flow plots showing the percentages of IFN-γ-expressing and IL9-expressing CD4+ (Th1 or Th9) and CD8+ (Tc1 or Tc9) CAR-T cells from spleen of treated mice at day 8 after transfer. **b** Summarized data (*n* = 5 mice) showing the frequency of IFN-γ-expressing CD4+ (Th1 or Th9; left panels) and CD8+ (Tc1 or Tc9; right panels) CAR-T cells in peripheral blood (upper panels) or spleen (lower panels) of treated mice at days 8 and 12 after transfer. **c** Representative flow plots showing the percentage of GrzB-expressing CD4+ (Th1 or Th9) and CD8+ (Tc1 or Tc9) CAR-T cells from spleen of treated mice at days 8 and 12 after transfer. **d** Summarized data (*n* = 5 mice) showing the frequency of GrzB-expressing CD4+ (Th1 or Th9; left panels) and CD8+ (Tc1 or Tc9; right panels) CAR-T cells in peripheral blood (upper panels) or spleen (lower panels) of treated mice at days 8 and 12 after transfer. **e** Representative flow plots showing the percentage of IL2-expressing CD4+ CAR-T cells from spleen of treated mice at days 8 and 12 after transfer. **f** Summarized data (*n* = 5 mice) showing the frequency of IL2-expressing CD4+ CAR-T cells in peripheral blood and spleen of treated mice at days 8 and 12 after transfer. Data are presented as mean ± SD. *$P < 0.05$ and ***$P < 0.001$, two-sided Student's *t*-test. Source data are provided as a Source Data file.

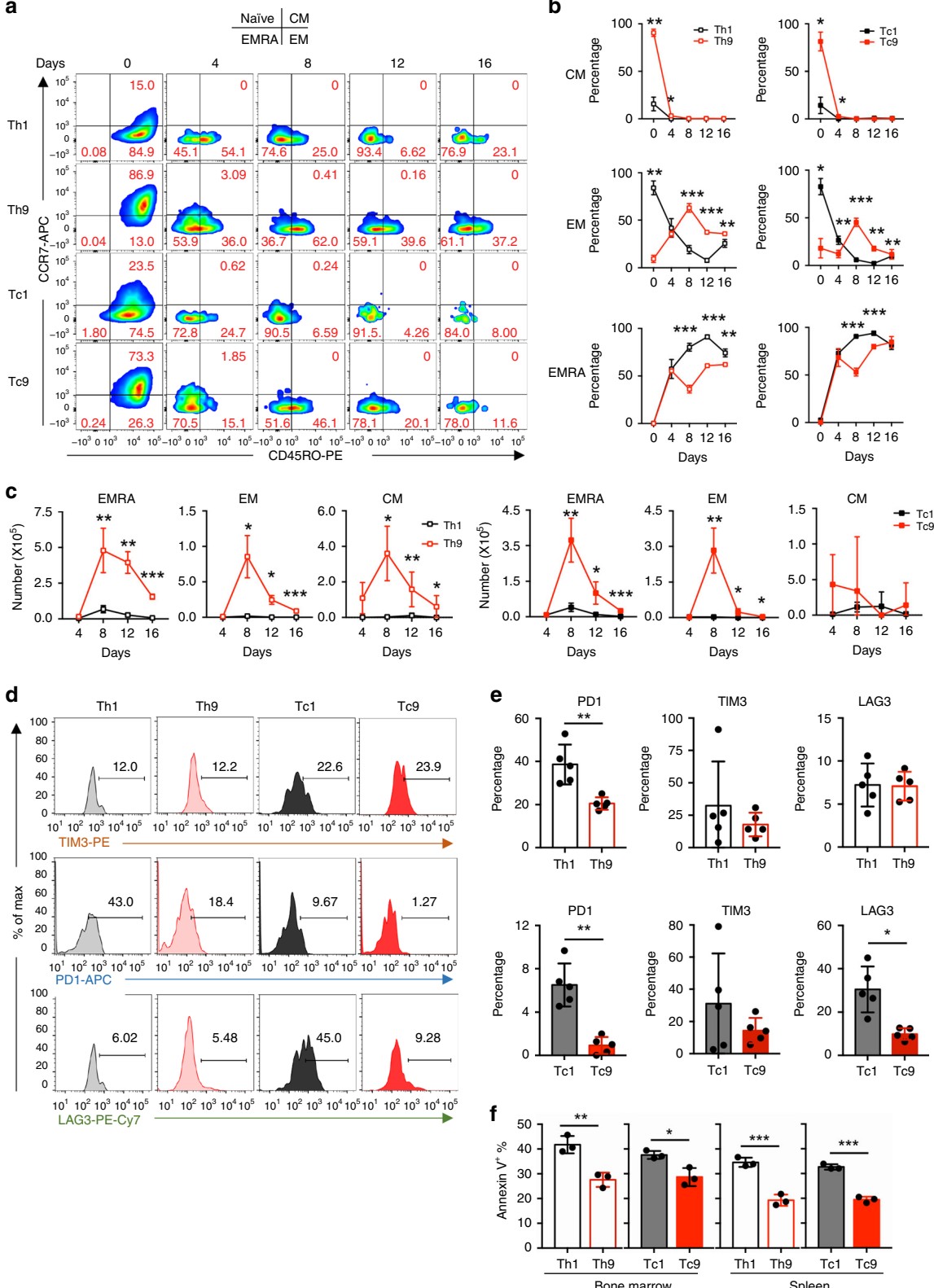

**Fig. 6 T9 CAR-T cells are long-lived and less-exhausted T cells in vivo. a** Representative flow plots showing the frequency of CD4+ or CD8+ CCD7+ and CD45R0+ CAR-T cells from spleen of treated mice at indicated days after CAR-T cell infusion. Cells were pre-gated for GFP+CD3+CD8+ or GFP+CD3+ CD4+ T cells. **b** Frequency and **c** total numbers of T cell subsets in CAR-T cells in spleen of treated mice at indicated days after CAR-T cell infusion (n = 4 mice at day 4, n = 5 mice at other days). **d** Representative flow plots showing expression of different exhaustion markers on CAR-T cells from spleen of treated mice at day 4 after CAR-T cell infusion. **e** Frequency of PD1+, TIM3+, and LAG3+ Th1 or Th9 (upper panels) or Tc1 or Tc9 (lower panels) CAR-T cells in spleen of treated mice at day 4 after CAR-T cell infusion (n = 5 mice). **f** Frequency of annexin V+ apoptotic CAR-T cells in bone marrow and spleen of treated mice at day 4 after transfer (n = 3 mice). Data are presented as mean ± SD. *P < 0.05 and ***P < 0.001, two-sided Student's t-test. Source data are provided as a Source Data file.

cultured with IL2. Our findings demonstrate that transfer of T9 CAR-T cells displays a greater efficacy to eradicate both liquid and solid tumors in xenografted mouse models compared to T1 CAR-T cells. Of note, other studies have shown that expanding CAR-T cells with addition of IL7 and IL15 is also more efficient than conventional method using IL2[34,35], because these cytokines could promote proliferation and survival of memory T cells and enhance immunotherapy efficacy of CAR-T cells[36].

We comprehensively profiled the transcriptomic state of T9 CAR-T cells and found that they displayed distinct cytokine expression profiles and reduced expression of exhaustion and terminal differentiation markers, which is largely aligned with

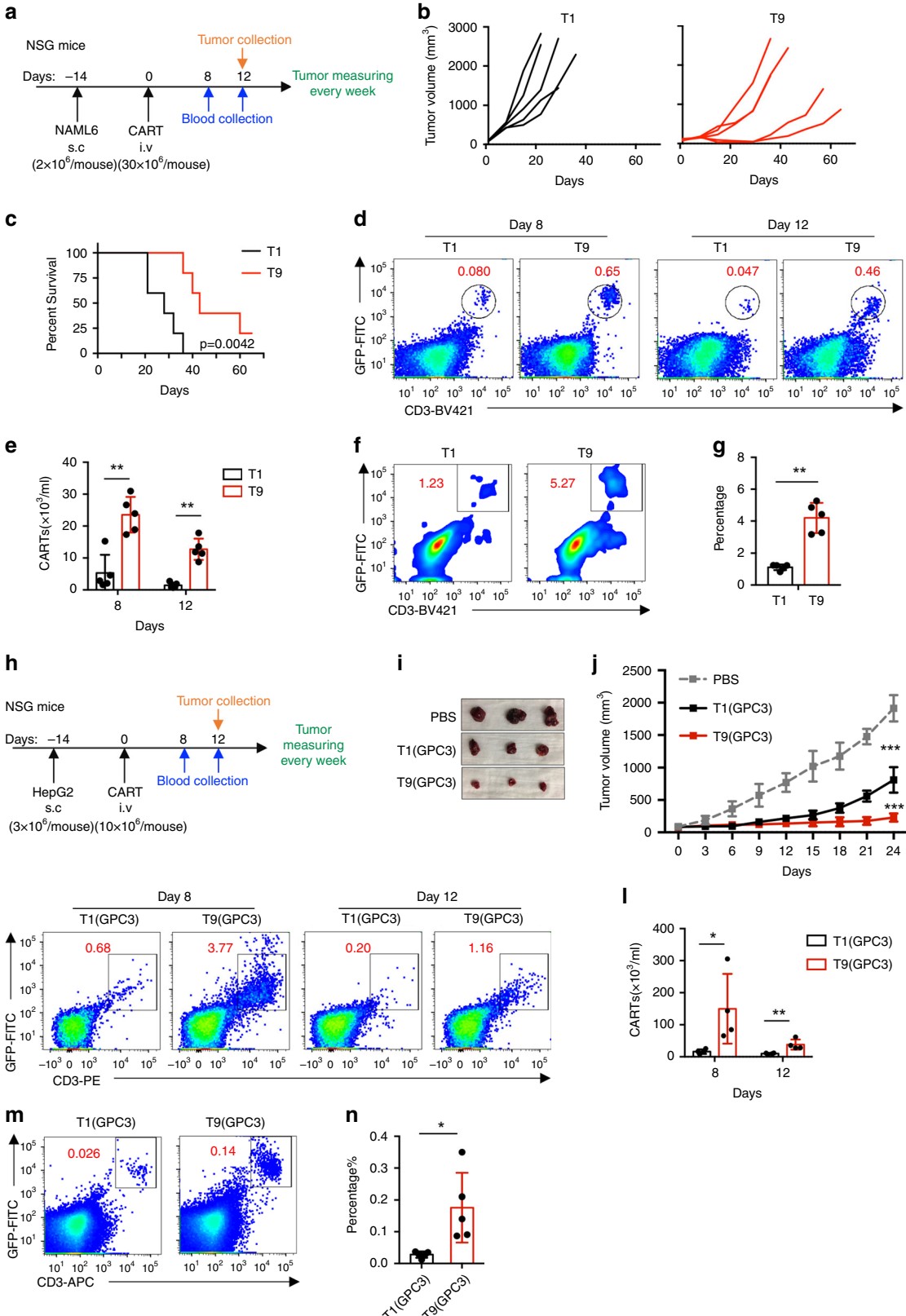

**Fig. 7 T9 CAR-T cells exert greater antitumor activity in solid tumor models. a** NSG mice were inoculated with $2 \times 10^6$ of NALM6 cells subcutaneously. After 14 days, mice were randomly assigned to three groups and infused intravenously with $30 \times 10^6$ CAR-T cells. **b** Tumor volume in mice treated with T1 or T9 CAR-T cells. **c** Kaplan–Meyer survival plots for CAR-T cell-treated mice ($n = 5$ mice). Exact $P$ values from log-rank test are shown for T1 versus T9 CAR-T cell-treated mice. **d** Representative flow plots showing frequency of CAR-T cells in peripheral blood of treated mice at indicated days after CAR-T infusion. **e** Percentage of CAR-T cells in peripheral blood of treated mice measured at days 8 and 12 after CAR-T cell infusion ($n = 5$ mice). **f** Representative flow plots showing the frequency of tumor-infiltrating CAR-T cells at day 12 after CAR-T cell infusion. **g** Percentage of tumor-infiltrating CAR-T cells at day 12 after CAR-T cell infusion ($n = 5$ mice). **h** NSG mice were inoculated with $3 \times 10^6$ of HepG2 cells subcutaneously. After 14 days, mice were randomly assigned to three groups and infused intravenously with $10 \times 10^6$ CAR-T cells. **i** Representative tumor images from mice treated with the indicated T cells or PBS on day 24. **j** Growth curve of HepG2 xenografted mice treated with the indicated T cells or PBS ($n = 5$ mice). **k** Flow plots (left panels) and **l** summarized data (right panels; $n = 4$ mice) showing frequency of CAR-T cells in peripheral blood of treated mice at indicated days after CAR-T infusion. **m**, **n** Percentage of tumor-infiltrating CAR-T cells at day 12 after CAR-T cell infusion ($n = 5$ mice). Data are presented as mean ± SD. $*P < 0.05$ and $***P < 0.001$, two-sided Student's $t$-test. Source data are provided as a Source Data file.

data generated from our mouse studies[20–23]. However, unlike murine Th9 cells that represent an effector subset[22], GSEA analysis suggested that human Th9 CAR-T cells are enriched in the central memory stage, whereas human Th1 CAR-T cells are enriched in effector memory stage. In line with these observations, flow cytometry analysis showed a higher frequency of central memory subset (CCR7$^+$CD45RO$^+$) in T9 CAR-T cells. Tc9 cells are central memory T cells, which may have contributed to their prolonged persistence, enhanced self-renew and survival, and better antitumor ability in vivo after adoptive transfer.

Another contributor of the prolonged persistence of T9 CAR-T cells is their hyperproliferative capacity. In vitro expansion showed that Th9 polarizing condition helps T cells acquire increased proliferative capacity, which is evident by the enrichment of molecular signatures related to G2M transition, DNA replication, and cell cycle checkpoint, and by decreased frequency of apoptosis in T9 CAR-T cells. After a second exposure to stimuli both in vitro and in vivo, T9 CAR-T cells also exhibited increased proliferative capacity compared to T1 CAR-T cells. This could not be contributed to the Th9-polarizing condition as the second stimulation happened without IL4 or TGF-β. It is plausible that T9 CAR-T cells are less susceptible to activation-induced cell death (AICD) than T1 CAR-T cells because we observed fewer apoptotic CAR-T cells in T9 CAR-T cell-treated mice compared to T1 CAR-T cell-treated mice, and because T9 CAR-T cells expressed lower levels of Fas and Fas ligand than T1 CAR-T cells. Fas and Fas ligand play an important role in the programmed death of CAR-T cells after prolonged exposure to tumor cells in vivo[15].

In contrast to the striking clinical efficacy of CAR-T cells in patients with hematologic malignancies, no success has been achieved in patients with solid tumors[8]. One of the major roadblocks that result in poor clinical outcomes for CAR-T cell therapy in solid tumors is due to poor migration and infiltration of CAR-T cells into tumors[37], and recent studies have shown that CAR-T cell tumor infiltration is associated with antitumor activity[38–40]. Therefore, improving CAR-T cell trafficking to solid tumors has become an urgent task for researchers. In our study, we observed a better antitumor response and significantly more tumor-infiltrating CAR-T cells in T9 CAR-T cell-treated mice bearing either liquid or solid tumors than mice treated with T1 CAR-T cells. These findings suggest that T9 CAR-T cells may have a better tumor infiltration capacity than T1 CAR-T cells and may be used to more efficiently treat patients with solid tumors.

In summary, our findings highlight a promising clinical potential of human IL9-secreting T cells for CAR-T cell immunotherapy for human cancers, attributed to their central memory phenotype and that they are less exhausted, hyperproliferative, and long-lived in vivo. Furthermore, these CAR-T cells exerted a great antitumor efficacy against both liquid and solid tumors in human xenografted mouse models in comparison with classically

polarized, IFN-γ-secreting CAR-T cells. These unique features of T9 CAR-T cells endow them with greater potential for clinical immunotherapy, which can lead to increased frequency of complete remission, high survival rates, and decreased number of patient relapse after treatment.

## Methods

**Cell lines.** All cell lines were obtained from ATCC. K562 and NALM6 cell lines were stably transduced with GFP and firefly luciferase and were cultured in complete media (RPMI supplemented with 10% FBS, 2 mM GlutaMAX, 100 U/mL penicillin, and 100 μg/mL streptomycin). HepG2 cells were maintained in Eagle's minimum essential medium supplemented with 10% FBS, 2 mM GlutaMAX, 100 U/mL penicillin, and 100 μg/mL streptomycin.

**CAR-T cell production.** FMC63 scFv-based chimeric immune receptor followed by 4-1BB and CD3 intracellular domain was constructed into lentiviral vector backbone constructs and GPC3 scFv-based CAR followed by 4-1BB and CD3 intracellular domain was constructed into retroviral vector backbone. In both vectors, CAR sequences are followed by 2A and GFP. 293T cells were seeded on 100-mm plates and co-transfected with lentiviral vector plasmid or retroviral vector plasmid with the packaging plamids and TransIT®-2020 Transfection Reagent (Mirus). Viral supernatants were harvested 60 h post transfection, concentrated by PEG-it virus precipitation kit (System Biosciences) and stocked at −80 °C for future use.

Human T cells were isolated from healthy volunteer donor blood following leukapheresis by negative selection. T cells were cultured in T Cell Expansion media (ThermoFisher) supplemented with 100 U/mL penicillin, 100 mg/mL streptomycin sulfate, and 2 mM GlutaMAX, and stimulated with human T-activator CD3/CD28 dynabeads (ThermoFisher) at a ratio of 3:1 (beads to cells). Twenty-four hours after activation, T cells were transduced with pseudovirus. To generate classical T1 or Th1/Tc1 CAR-T cells, human CD3$^+$, CD4$^+$, or CD8$^+$ T cells were cultured with recombinant IL2 (Peprotech) added every other day at 100 IU/mL final concentration and cell density of $0.5 \times 10^6$–$1 \times 10^6$ cells/mL. To generate T9 CAR-T cells, human CD3$^+$ T cells were cultured in the presence of IL4 (10 ng/mL, R&D), TGF-β (1 ng/mL, R&D), anti-IFN-γ antibody (10 μg/mL), and IL2 as described above. CAR-T cells were harvested and used for in vitro assays or adoptive transfer experiments at days 14 or 16 since initial stimulation. Human T cells from healthy donors were purchased from Gulf Coast Regional Blood Center in Houston. All experiments complied with protocols were approved by Institutional Review Board at the Lerner Research Institute of Cleveland Clinic and the Houston Methodist Research Institute.

**Cytokine production assay.** Sorted CD4$^+$ or CD8$^+$ CAR-T cells ($2 \times 10^5$) and $1 \times 10^5$ tumor cells were cocultured in 96-well flat bottom plates for 24 h. Culture supernatants were collected and analyzed with BD Cytometric Bead Assay Kit (BD Bioscience) according to the manufacturer's instruction.

**Flow cytometry.** Immunophenotype of T cells was performed using standard staining and flow cytometry techniques as described before[41,42]. Briefly, combinations of fluorophore-conjugated anti-human monoclonal antibodies specific for CD3 (BioLegend, OKT3, Cat#317344, 1:500), CD4 (BioLegend, OKT4, Cat#317438, 1:500), CD8 (BioLegend, SK1, Cat#344724, 1:500), CCR7 (BioLegend, G043H7, Cat#353214, 1:500), CD45RO (BioLegend, UCHL1, Cat#304244, 1:500), PD1 (eBioscience, EBIOJ105, Cat#12-2799-42, 1:500), TIM3 (BioLegend, F38-2E2, Cat#345006, 1:500), LAG3 (eBioscience, 3DS223H, Cat#15-2239-42, 1:500) were used to label T cells in flow cytometry staining buffer for 30 min on ice after Fc blocking, followed by intracellular cytokine staining with the BD Fixation/Permeabilization Solution Kit. Data were acquired on LSRFortessa (BD Biosciences) and analyzed with FlowJo software (Treestar). Cytokine antibodies included IFN-γ (eBioscience, B27, Cat#MHCIFG04, 1:500), IL9 (BioLegend, MH9A4, Cat#507614,

1:500), IL2 (BioLegend, MQ1-17H12, Cat#500342, 1:500), GrzB (BD Biosciences, GB11, Cat#561142, 1:500), or annexin V (BD Bioscience, Cat#550475, 1:50).

**Luciferase assay**. Cytotoxicity of CD19 CAR-T cells was measured by Luciferase assay. Briefly, ffLuc expressing target cells (K562, CD19 expressing K562, and NALM6) were resuspended in RPMI medium supplemented with 10% FBS in 96-well tissue culture plates. CAR-T cells were added at varying effector to target cell ratios. After 24-h incubation, cells were lysed in lysis reagent (Promega). Luminescence of the lysates was analyzed using a plated spectrophotometer. Spontaneous release (no CAR-T cells added) and maximum release (treated with lysis reagent) were set up. The percentage of specific lysis was calculated according to the standard formula: % specific lysis = $100 \times$(experimental release–spontaneous release)/(maximum release–spontaneous release).

A flow cytometry-based cytotoxicity assay was used to determine cytotoxicity of GPC3 CAR-T cells. Briefly, tumor cells were pre-stained with CellTrace™ Far Red (Thermofisher) and cocultured with CAR-T cells for 24 h. After that, cultures were added with counting beads (Thermofisher) for normalization prior to flow cytometry analysis.

**Real-time PCR**. qPCR was performed as described by He J et al. [43]. RNA isolations were done with RNeasy kit (QIAGEN) and single-strand cDNA was synthesized with High Capacity cDNA Reverse Transcription Kit (Applied Biosystems). Primers were purchased from Applied Biosystems and real-time PCR was performed using the Taqman method with an Applied Biosystems 7500 sequence detection system. Expression of mRNA for genes of interest was analyzed by SYBR green real-time PCR normalized to the expression of the housekeeping gene gapdh.

**Proliferation assay**. T cells were harvested at day 14 after initial stimulation, and stained using CellTrace™ Far Red proliferation kit (Invitrogen) according to manufacturer instruction. Far Red-labeled cells were cultured in the presence of IL2 at 100 IU/mL final concentration. Cells were harvested, stained with anti-CD3, CD4, or CD8 antibodies, and measured by flow cytometry 3 days after stimulation.

For evaluating proliferative capacity, CAR-T cells were cocultured with tumor cells or beads at a ratio of 1:1 (beads to cells) for 4 days, followed by staining with anti-CD3, CD4, or CD8 antibodies, and then quantified using CountBright™ absolute counting beads (ThermoFisher) on a BD Fortessa flow cytometer.

**Metabolism assays**. To measure oxygen consumption rates (OCR) and extracellular acidification rates (ECAR), T cells were suspended in XF media (non-buffered RPMI 1640 containing 25 mM glucose, 2 mM L-glutamine, and 1 mM sodium pyruvate) and incubated in standard culture condition for 60 min, then switched to a $CO_2$-free incubator for another 30 min, followed by measuring under basal conditions and in response to 1.5 μM oligomycin, 1.5 μM fluoro-carbonyl cyanide phenylhydrazone (FCCP), 50 nM rotenone, and 1 μM antimycin A (Sigma) using XF-24 Extracellular Flux Analyzer (Seahorse Bioscience)[44].

**RNASeq analysis**. On day 16 of culture total RNA was isolated from CAR-T cells using RNeasy Mini kits (Qiagen) following the manufacturer's instruction. Quality of total RNA was evaluated using RNA 6000 Nano LabChip (Agilent 2100 Bioanalyzer, Santa Clara, CA). All samples had intact 18S and 28S ribosomal RNA bands with RIN numbers from 8.1 to 10 and RNA 260/280 ratios between 1.9 to 2.0. Raw data were processed with SOAPnuke, Bowtie, RSEM, and differentially expressed genes were identified using DEseq2 algorithms and visualized with gplots package in R.

GSEA was run for each T-cell subset in pre-ranked list mode with 1000 permutations (nominal P-value cutoff < 0.05). The T cell signature gene sets (down and up) from the Broad Institute Molecular Signature Database were used:
    KEGG_APOPTOSIS
    GO_CELL_CYCLE_CHECKPOINT
    GO_CELL_CYCLE_DNA_REPLICATION
    GO_REGULATION_OF_CELL_CYCLE_G2_M_PHASE_TRANSITION
    GSE11057_EFF_MEM_VS_CENT_MEM_CD4_TCELL_DN
    GSE26928_EFF_MEM_VS_CENTR_MEM_CD4_TCELL_DN
    GSE11057_EFF_MEM_VS_CENT_MEM_CD4_TCELL_UP
    GSE26928_EFF_MEM_VS_CENTR_MEM_CD4_TCELL_UP
For analysis of central memory subset, signature gene sets are from merged GSE11057_EFF_MEM_VS_CENT_MEM_CD4_TCELL_DN;
    and GSE26928_EFF_MEM_VS_CENTR_MEM_CD4_TCELL_DN; For analysis of effector memory subset, signature gene sets are from merged GSE11057_EFF_MEM_VS_CENT_MEM_CD4_TCELL_UP;
    and GSE26928_EFF_MEM_VS_CENTR_MEM_CD4_TCELL_UP.

**In vivo xenograft mouse studies**. Human NALM6 leukemia xenografted mouse model was used to examine the antitumor activity of human CD19 CAR-T cells. Briefly, immunodeficient NSG mice (Jackson Laboratory, ME) at 8 weeks of age were inoculated intravenously with $1 \times 10^6$ NALM6-Luc cells. Seven days later, $4 \times 10^6$ CD19 CAR-T cells were adoptively infused via tail vein injection. Tumor burdens were evaluated weekly by bioluminescence on IVIS imaging system

(Xenogen Corp.). Survival was monitored. Mice were sacrificed when they develop signs of hind limb paralysis[45]. Mice were housed in a clean facility in 12/12 light/dark cycle, with an ambient temperature of 65–75 °F and 40–60 humidity.

For subcutaneous mouse model, 8–10-week-old NSG mice were injected subcutaneously with $2 \times 10^6$ NALM6 cells. Two weeks after inoculation, $30 \times 10^6$ CAR-T cells were injected intravenously into tumor-bearing mice. Tumor sizes were measured every week. For liver cancer subcutaneous mouse model, NSG mice were injected subcutaneously with $3 \times 10^6$ HepG2 cells. Two weeks after inoculation, $10 \times 10^6$ CAR-T cells were injected intravenously into tumor-bearing mice. All mice were maintained in American Association of Laboratory Animal Care-accredited facilities, and the studies were approved by the Institutional Animal Care and Use Committee of the Lerner Research Institute of Cleveland Clinic and the Houston Methodist Research Institute.

Mouse blood (50 μL) was collected via tail blood collection at the indicated time points. After lysing using BD Lysing Solution, blood cells were stained with the indicated antibodies and enumerated using CountBright™ absolute counting beads (ThermoFisher) on a BD Fortessa flow cytometer. Spleen total cells were enumerated using trypan blue exclusion. Absolute numbers of transferred CAR-T cells in the spleen of treated mice were calculated by multiplying the frequency of indicated positive cells by the total number of viable cells. Bone marrow cells were isolated from both femurs and tibias of treated mice.

**Statistical analyses**. All statistical calculations and graphs were generated by GraphPad Prism 7.04. For statistical analysis, Student's t-test was used. A P-value < 0.05 was considered statistically significant. Results are presented as mean ± SD unless otherwise indicated. Mouse survival was analyzed using the long-rank test.

**Reporting summary**. Further information on research design is available in the Nature Research Reporting Summary linked to this article.

## Data availability
RNA-Seq data are available at NCBI's functional genomic data repository Gene Expression Omnibus (GEO) under accession code GSE156075. Gene set for GSEA analysis including KEGG_APOPTOSIS, GO_CELL_CYCLE_CHECKPOINT, GO_CELL_CYCLE_DNA_REPLICATION, GO_REGULATION_OF_CELL_CYCLE_G2_M_PHASE_TRANSITION, GSE11057_EFF_MEM_VS_CENT_MEM_CD4_TCELL_DN, GSE26928_EFF_MEM_VS_CENTR_MEM_CD4_TCELL_DN, GSE11057_EFF_MEM_VS_CENT_MEM_CD4_TCELL_UP, GSE26928_EFF_MEM_VS_CENTR_MEM_CD4_TCELL_UP, are available from The Molecular Signature Database (MSiDB). All the other data supporting the findings of this study are available within the article, supplementary information, source files, and from the corresponding author upon reasonable request. Source data are provided with this paper.

## Code availability
Our samples were sequenced and analyzed by BGI Americas Company. The Company only provided software information used in the analysis process, but not the program information or scripts due to copyright issues involved. Scrip for data visualization analysis is open-source and available from the Liulintao GitHub repository (https://github.com/Liulintao).

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

## Acknowledgements

This work was supported by NCI R01 CA200539 and Cancer Prevention & Research Institute of Texas Recruitment of Established Investigator Award (RR180044). Q.Y. and his research group are also supported by NCI R01s (CA211073, CA214811, and CA239255).

## Author contributions

L.L. and Q.Y. conceptualized the study and designed the experiments. L.L., X.M., and Q.Y. wrote the paper. L.L., E.B., and X.M. performed the majority of experiments and statistical analyses. W.X. performed some of animal studies. J.Q., L.Y., P.S., Q.W., L.X., M.Y., and Y.L. provided critical suggestions.

## Competing interests

The authors declare no competing interests.
