## [Peer Review File · Nature Communications]

REVIEWER COMMENTS

Reviewer #1 (Remarks to the Author): with expertise in Tc/Th9 cells and anti-tumor immune response

CAR-T cell therapy are a new and important therapy in B cells leukemia. The authors may the very interesting discovery that CD19-targeted human CAR-T cells polarized and expanded under a Th9-polarized condition have enhanced anti-tumor activity against established leukemia.

These Th9 CART are more efficient than classical CART. Th9 CAR-T cells secreted IL9 expressed central memory phenotype and are not exhausted. They seem to proliferate more than classical CAR T and have less apoptotic machinery. In vivo they differentiate in more effector cells that express IFN γ .

The concept is very interesting, an important in a clinical point of view. However I am not convinced by the control arm and the mechanism why is not completely addressed.

In detail.

- CART developed in IL2 medium are not Th1 cells. Authors need to generate CART with IL12 to induce Th1 cells.

- Probably a more complete comparison using either Th17 or Th2 will importantly improve the manuscript.

- Th9 carT seems to not express PU1. It is important to better address this issue because PU1 is one of the more specific transcriptional factors involved in Th9 differentiation.

- In such artificial in vivo model we do not know which cells and which molecules are effectors in vivo. What is the effect of GRanzyme B invalidation? What is the effect of IL9 or IFN γ invalidation?

- More importantly the therapy consists in a mix of CD4 and CD8 T cells which cells are therapeutic? We can suspect that CD8 are more cytotoxic and are major actors of the therapeutic effect? but it is just a hypothesis and an alternative hypothesis could be proposed and must be experimentally addressed.

- In a more general conceptual manner it seems that Th9 CAR T cells are less differentiated and switch differentiation in vivo. Such data are important and must be better addressed. What is the effect of Th9 polarization on the exhaustion program (TOX, TCF, SlamF6 and so on...)? Does TGF or IL4 are involved in this tuning of the exhaustion program...? Does the complete Th9 program is important in this reduction of the exhaustion process or only a part of this program driven by IL4 or TGF is sufficient to reduce exhaustion and promote efficacy of CAR T?

Reviewer #2 (Remarks to the Author): with expertise in transcriptomic and T cell differentiation

The study by Liu and colleagues examines the utility of CAR-T cell therapy using Th9 stimulation conditions to expand cells for transduction. They show that activation of T cells under Th9 conditions generate CAR-T cells that have greater self-renewal capacity, enhanced persistence/survival after in vivo transfer in an animal model, and are still effective at control of model tumour challenge.

This is a lovely study. The data largely support the conclusions with experiments well done and presented. This addresses a key question and problem in the area about how to perhaps target prolonged persistence of CAR-T cells in the long term, as well as important pre-clinical data showing effectiveness against solid tumors.

I do have a couple of issues that, if addressed, I would like to think help clarify some issues that popped into my mind whilst reading the study.

The first are factors that are directly contributing to the greater survival/self-renewal capacity of

TH9 generated CAR-T cells. The authors focus on obvious candidates such as anti-apoptosis and cell cycle regulators but it would be interesting to know whether these end stage markers correlate with increased expression/frequency of factors that direct self renewal/quiescence. In particular, do the authors have any data about the expression of Tcf7, FOXO1, Lef1, VISTA, Btg1/2, mTOR? One might assume these could be examined in the RNA-seq data. In line with this, more detail regarding the genes associated with TCM would be of interest, particularly more explanation of how overrepresentation of these transcripts regulate persistence (eg MCM, CDK2).

I was a bit confused about what the epigenetic hall markers actually are (pg 8, line 153). Are the authors references particular transcripts over represented? What are they, how would they contribute to the observed phenotype?

In the results, it would be interesting to outline the precise timelines in the text being examined.

Line 187, page 10. This sentence is incomplete, not clear what is mature?

In the discussion, it might be worth highlighting recent literature that suggests that IL-15 actually promotes proliferation and survival of virtual memory T cells. This might help explain deficiencies in CAR T cell therapy given it is assumed they are : "naive" but are in fact over represented as we age. Moreover, with ageing, they become more dysfunctional but are still responsive to IL-15 (Quinn et al., PMID: 29924995).

The hyperproliferative response of the TH9 CAR T cells is a clear explanation for improved outcome after tumour challenge. This was accompanied by clear differentiation into effector T cells. One aspect that would have been good to learn more about is whether these capacity for enhanced differentiation was also related to the capacity to form TRM memory precursors. Examination of CD69, CD103 and Hobit might provided some indication, particularly in the solid tumor model (or are there are chemokine/migratory receptors that might help explain enhanced migration/persistence in solid tumors?).

Reviewer #3 (Remarks to the Author): with expertise in CAR-T cells

There's a clear need to improve outcome of CAR T-cell treated patients, as loss of persistence is seen in multiple trials, related to intrinsic CAR problems as well as other (AICD, immune response). Adaptations of the expansion process may indeed benefit the product and clinical outcome of patients.

General comments:

The overall work is interesting and well performed. Its main limitation is studying only one type of CAR. Proof that the T9 culture conditions is superior should be shown beyond an FMC63-41BB-Z CAR. For example – showing benefit in CD28-costimulated CARs, or with different ScFVs, would be of essence.

Also – targeting solid tumors is mentioned and discussed in the paper, but this was not truly attempted. A NALM6 model subcutaneously injected to an NSG mouse is perhaps a lymphoma model but certainly not a true solid tumor as far as the microenvironment, let alone in an immune-deficient animal.

Major specific comments:

Were the Tc9 / Th9 grown separately or in bulk ? and control Tc1/Th1 T-cells ? If so – how were they separated ?

What was the transduction efficacy of the Tc9/Th9?

Specifically - in figure 1c-d, flow cytometry shows that only a subset of cells express IL9. Can you confirm these were the CAR-positive cells (by flow for example)?

This is even more important given the in-vivo data (fig 5a-b, line 176-178), showing lack of IL9 expression in the cells in vivo. An alternative explanation would be that the IL9+ cells die in vivo and the 'regular' CAR+ T-cells persist.

Minor comments:

In the introduction - the overall survival in ALL treated with CAR T cells is not 1 year as stated, perhaps the median OS (line 43-45); also, the 30% EFS/OS long-term is with a short-durability CD28-based CAR. Data is lacking regarding long term results of 41BB-costimulated CAR T cells, known to have longer persistence. For DLBCL, seems that long-term outcome is ~40% EFS (Locke, Lancet Oncol 2018).

In figure 1, where is the gate in the IFN γ flow-plot drawn? Can dot plots also be shown for IFN γ in 1d (such as in 1c for IL9) ?

The regular CAR T (T1CAR) cells seem to have increased EM phenotype (fig 3) despite having a 41BB co-stimulatory domain (these percentages seems to be appropriate to CD28-based CARs), unlike what is seen from different groups. Increase in mitochondrial mass with T9 CAR is important. However, would be cautious stating this is convincing of a less-differentiated state (line 202) but could be

In-vivo model -can you comment on the cause of relapse in these mice ? Loss of persistence

Typos-

Many typos throughout of NALM6 (correct) vs NAML6 - please revise

Would recommend changing the beginning of the discussion, as age of the patient/donor is not a factor, and, for example, age of lymphocyte donors to this study was at all not mentioned.

Please revise reference style as some are written differently (abstract format ?) and some references are written with little justification (e.g. ref#3 Kantarjian et al which has no relation to CAR therapy mentioned in this sentence) .

We would like to thank the reviewers for their thoughtful and constructive criticisms. We believe that with their helpful suggestions, this revised manuscript is substantially improved over the previous submission. All revisions are underlined in the revised manuscript. Below are point-by-point responses to the comments.

Reviewers' Comments:

Reviewer #1:

with expertise in Tc/Th9 cells and anti-tumor immune response.

CAR-T cell therapy are a new and important therapy in B cells leukemia. The authors may the very interesting discovery that CD19-targeted human CAR-T cells polarized and expanded under a Th9-polarized condition have enhanced anti-tumor activity against established leukemia. These Th9 CART are more efficient than classical CART. T9 CAR-T cells secreted IL9 expressed central memory phenotype and are not exhausted. They seem to proliferate more than classical CAR T and have less apoptotic machinery. In vivo they differentiate in more effector cells that express IFN γ . The concept is very interesting, an important in a clinical point of view. However, I am not convinced by the control arm and the mechanism why is not completely addressed.

In detail, CART developed in IL2 medium are not Th1 cells. Authors need to generate CART with IL12 to induce Th1 cells.

Response: We understand the concern. The goal of our study was to compare antitumor efficacy of T9 CAR-T cells with IL2-activated CAR-T cells because all FDA-approved, currently used CAR-T cells are IL2-activated.

However, to address the concern, we generated CAR-T cells in medium with (T1-IL12) or without IL12 (T1) in vitro and compared their phenotype and function. We found that although IL12 greatly enhanced IFN γ expression in T1-IL12 compared to T1 CAR-T cells (Figure A-a, below), there was no significant difference in their proliferation (Figure A-b) or cytotoxicity (Figure A-c). As T cell differentiation state is associated with their in vivo anti-tumor efficacy, we compared their CCR7 and CD45RO expression, which could be used to differentiate naïve (CD45RA⁺CCR7⁺), central memory (CD45RO⁺CCR7⁺), effector memory (CD45RO⁺CCR7⁻), and terminally differentiated effector memory (CD45RO⁻CCR7⁻) T-cell subpopulations. Results indicate that IL12 did not change T cell differentiation state (Figure A-d). However, IL12 significantly increased expression of exhaustion markers such as LAG3 (Figure A-e), PD1 and TIM3 (Figure A-f), as well as PD1⁺TIM3⁺ cells (Figure A-f). The data suggest that, similar to IL2-activated cells, IL12-activated CAR-T cells are also exhausted.

Figure A. IL12 induces CAR-T cell exhaustion in vitro. CAR-T cells were generated in culture with or without IL12. At day 14, T cells were subject to IFN γ staining (a), proliferation by trypan blue exclusion (b), cytotoxicity against NALM6 tumor cells by luciferase assay in 24-hour culture (c), memory marker (CCR7 and CD45RO) staining (d), and exhaustion marker staining, including LAG3 (e), TIM3 and PD1 (f). For flow analysis, cells were pre-gated for GFP⁺CD3⁺. **P < 0.01.

Probably a more complete comparison using either Th17 or Th2 will importantly improve the manuscript.

Response: We appreciate the suggestion. Our previous studies have shown that tumor-specific Th9 cells are better effector T cells in comparison with Th1 or Th17 cells to eradicate established tumors in vivo (Lu et al, JCI 122:4160–4171, 2012, Lu et al, Cancer Cell 33:1048-1060, 2018), which prompted us to perform the preclinical study to explore whether human T9 CAR-T cells may be better effector T cells than the currently used, IL2-activated CAR-T cells for cancer treatment. As Th2 cells are not considered effector T cells, we have not used these cells in our cancer immunotherapy studies.

T9 CAR T seems to no express PU1. It is important to better address this issue because PU1 in one of the more specific transcriptional factors involve in Th9 differentiation.

Response: We understand the concern, rechecked the expression of PU.1 in the cells, and confirmed the result. PU.1 and also many other transcription factors have been reported to be involved in Th9 cell differentiation (Semin Immunopathol. 39: 11–20, 2017). We reported that FOXO1 is a transcription factor for Th9 cell differentiation (Bi et al, *Science Signaling* 10; eaak9741, 2017). However, no established master transcription factor for Th9 cell differentiation has been identified so far. Research by Xiao et al. also showed that under certain circumstances,

PU.1 (encoded by *Sfp1*) is not involved in Th9 differentiation. They demonstrated that OX40 is a powerful inducer of Th9 cells in vitro, and *Sfp1* deletion did not alter IL9 expression induced by OX40 (Nat Immunol 13: 981–990, 2012). Therefore, it is not surprising that PU.1 expression is low in our cells.

IN such artificial in vivo model we do not know which cells and which molecules are effector in vivo. What is the effect of Granzyme B invalidation? What is the effect of IL9 or IFN γ invalidation?

Response: We appreciate the suggestions. First, our data (Fig. 5C and 5D) showed that there were no significant differences in GzmB production between in vivo transferred Tc1 and Tc9 cells. Second, our new data in new Fig. S7 show that CD8⁺ CAR-T cells exhibited superior antitumor efficacy compared to CD4⁺ CAR-T cells, suggesting that GzmB may not play a major role in T9 CAR-T cell effector function.

To examine the effect of IL9 or IFN γ on the efficacy of T9 CAR-T cells, we treated NAML6-inoculated mice with their neutralizing antibodies and injected with T9 CAR-T cells (new Fig. S8). Anti-IL9 or anti-IFN γ did not significantly affect tumor growth and survival of tumor-bearing mice receiving T9 CAR-T cell treatment. Moreover, these antibodies did not significantly affect the persistence of T9 CAR T cells in vivo. The results demonstrate that IL9 and IFN γ are not major contributors for T9 CAR T cell-mediated antitumor efficacy. These results are also consistent with our previous study that OT-II Th9 cells did not require IL9 or IFN γ to exert their antitumor activity in mice (Lu et al, Cancer Cell 33:1048-1060, 2018).

More importantly the therapy consists in a mixed of CD4 and CD8 T cells which cells are therapeutic? We can suspect that CD8 are more cytotoxic and are major actor of the therapeutic effect? but it is just a hypothesis and alternative hypothesis could be proposed an must be experimentally addressed.

Response: We agree with the comments and conducted new experiments to address the question. Accordingly, we isolated human CD4⁺ and CD8⁺ T cells, polarized them in vitro under Th9- or Th1 condition, and injected them to tumor-bearing mice (new Fig. S7a). Our results show that Tc9 CAR-T cells exhibited superior antitumor efficacy compared to Th9, Th1, or Tc1 CAR-T cells (new Fig. S7b-d), and Th9 CAR-T cells had similar antitumor ability as Tc1 cells in vivo. In addition, Th9 or Tc9 CAR-T cells showed stronger proliferation ability than Th1 and Tc1 CAR-T cells (new Fig. S7e).

In a more general conceptual manner it seems that Th9 CAR T cells are less differentiated and switch differentiation in vivo. such data are important and must be better addressed. What is the effect of Th9 polarization on exhaustion program (TOX, TCF, SlamF6 and so on...) Does TGF or IL4 are involved in this tuning of exhaustion program... Does the complete Th9 program is important in this reduction of exhaustion process or only a part of this program drive by IL4 or TGF is sufficient to reduce exhaustion and Promote efficacy of CAR T.

Response: We appreciate the comments. We re-analyzed the expression of exhaustion-related genes (*Tcf3*(E2A), JUN, FOXO1, BTG2, mTOR, ID2, Slamf6, TCF1(TCF7), VISTA, BTG1,

Lef1, TOX) in our RNA-seq data and found that only the gene JUN, which was found to induce exhaustion resistance in CAR-T cells (Nature 576, 293–300, 2019), was highly expressed in Th9 CAR-T cells (new Fig. S5a). To confirm this result, Th9 CAR-T cells expanded at day 14 were subjected to western blot analysis, which showed that Th9 cells exhibited increased c-jun expression and c-jun phosphorylation at Ser73 compared to Th1 cells (new Fig. S5b).

Next, to determine which cytokine plays a major role in shaping the properties of T9 CAR-T cells, we polarized CAR-T cells under 4 different conditions (none, IL4, TGF β , IL4+TGF β). IL2 was added to all of the culture to support T cell survival and growth. Results showed that TGF β alone could induce the expression and phosphorylation of c-jun (new Fig. S5c) and expression of CCR7 (new Fig. S5d). However, TGF β alone greatly suppressed cell growth, which could be rescued by addition of IL4 (new Fig. S5e). In addition, IL4 reduced the percentage of PD1⁺TIM3⁺ population (new Fig. S5f). Overall, the properties of T9 CAR-T cells, including IL9 expression, is the result of a combined effect of TGF β and IL4 (new Fig. S5g).

Reviewer #2

with expertise in transcriptomic and T cell differentiation

The study by Liu and colleagues examines the utility of CAR-T cell therapy using TH9 stimulation conditions to expand cells for transduction. They show that activation of T cells under TH9 conditions generate CAR-T cells that have greater self-renewal capacity, enhanced persistence/survival after in vivo transfer in an animal model, and are still effective at control of model tumour challenge. This is a lovely study. The data largely support the conclusions with experiments well done and presented. This addresses a key question and problem in the area about how to perhaps target prolonged persistence of CAR-T cells in the long term, as well important pre-clinical data showing effectiveness against solid tumors. I do have a couple of issues that, if addressed, I would like to think help clarify some issues that popped into my mind whilst reading the study.

The first are factors that are directly contributing to the greater survival/self renewal capacity of TH9 generated CAR-T cells. The authors focus on obvious candidates such as anti-apoptosis and cell cycle regulators but it would be interesting to know whether these end stage markers correlate with increased expression/frequency of factors that direct self renewal/quiescence. In particular, do the authors have any data about the expression of Tcf7, FOXO1, Lef1, VISTA, Btg1/2, mTOR? One might assume these could be examined in the RNA-seq data. In line with this, more detail regarding the genes associated with TCM would be of interest, particularly more explanation of how overrepresentation of these transcripts regulate persistence (eg MCM, CDK2).

Response: We appreciate the comments. We re-analyzed the expression of exhaustion-related genes (Tcf3(E2A), JUN, FOXO1, BTG2, mTOR, ID2, Slamf6, TCF1(TCF7), VISTA, BTG1, Lef1, TOX) in our RNA-Seq data and found that only the gene JUN, which was found to induce exhaustion resistance in CAR-T cells (Nature, 576, 293–300, 2019), was highly expressed in Th9 CAR-T cells (new Fig. S5a). To confirm this result, Th9 CAR-T cells expanded at day 14 were

subjected to western blot analysis, which showed that Th9 cells exhibited increased c-jun expression and c-jun phosphorylation at Ser73 compared to Th1 cells (new Fig. S5b).

c-Jun is essential for transition beyond the G1/S and G2/M checkpoints (Blood 79, 2107-2115) and c-Jun transactivates a large number of genes that are directly or indirectly involved in controlling cell proliferation (Nature Reviews Cancer 3, 859–868, 2003), which might explain the enrichment of proliferation- and cell cycle-related genes in T9 CAR-T cells. In addition, a recent study uncovered that c-Jun could enhance expansion potential, diminish terminal differentiation, and improve antitumor efficacy of CAR-T cells (Nature 576, 293–300, 2019). Therefore, T9 CAR-T cell's features, which include hyperproliferation and less exhaustion, might be explained by their enhanced c-Jun expression level.

I was a bit confused about what the epigenetic hall markers actually are (pg. 8, line 153). Are the authors references particular transcripts over represented? What are they, how would they contribute to the observed phenotype?

Response: Sorry for the confusion. We have rephrased the sentence to make it clear.

In the results, it would be interesting to outline the precise timelines in the text being examined.

Response: Thank you for the suggestion. We have re-outlined the timelines in Results and Figure Legends.

Line 187, page 10. This sentence is incomplete, not clear what is mature?

Response: Thank you for the comment. We have changed the title accordingly.

In the discussion, it might be worth highlighting recent literature that suggests that IL-15 actually promotes proliferation and survival of virtual memory T cells. This might help explain deficiencies in CAR T cell therapy given it is assumed they are:"naive" but are in fact over represented as we age. Moreover, with ageing, they become more dysfunctional but are still responsive to IL-15 (Quinn et al., PMID: 29924995).

Response: Thank you for the suggestion. Accordingly, we have now highlighted it in the first paragraph of Discussion.

The hyperproliferative response of the TH9 CAR T cells is a clear explanation for improved outcome after tumor challenge. This was accompanied by clear differentiation into effector T cells. One aspect that would have been good to learn more about is whether these capacity for enhanced differentiation was also related to the capacity to form TRM memory precursors. Examination of CD69, CD103 and Hobit might provide some indication, particularly in the solid tumor model (or are there are chemokine/migratory receptors that might help explain enhanced migration/persistence in solid tumors?).

Response: We appreciate the suggestions. We analyzed several markers on T9 CAR-T cells and found that T9 CAR-T cells exhibited higher CD69 expression than T1 CAR-T cells, which could help explain the less exhausted and less differentiated phenotype of T9 CAR-T cells (Figure B, below). In addition, our RNA-seq data (Fig. 1f) showed that Th9 CAR-T cells expressed higher level of CXCR4 compared to Th1 CAR-T cells, which could facilitate T cell trafficking to the bone marrow (Blood 125: 2087–2094, 2015, Eur J Immunol. 49:576-589, 2019). These findings may help explain why T9 CAR-T cells have stronger tumor infiltrating ability.

Figure B. Th9 CAR-T cells show higher expression of CD69 than Th1 CAR-T cells. CAR-T cells were differentiated with anti-CD3/CD28 beads for 14 days, and CD69 expression was analyzed by flow cytometry. *P < 0.05, **P < 0.01.

Reviewer #3:

with expertise in CAR-T cells

There's a clear need to improve outcome of CAR T-cell treated patients, as loss of persistence is seen in multiple trials, related to intrinsic CAR problems as well as other (AICD, immune response). Adaptations of the expansion process may indeed benefit the product and clinical outcome of patients.

General comments:

The overall work is interesting and well performed. Its main limitation is studying only one type of CAR. Proof that the T9 culture conditions is superior should be shown beyond an FMC63-41BB-Z CAR. For example – showing benefit in CD28-costimulated CARs, or with different ScFVs, would be of essence. Also – targeting solid tumors is mentioned and discussed in the paper, but this was not truly attempted. A NALM6 model subcutaneously injected to an NSG mouse is perhaps a lymphoma model but certainly not a true solid tumor as far as the microenvironment, let alone in an immune-deficient animal.

Response: We appreciate the comments. To prove that T9 culture condition is superior beyond a FMC63-41BB-Z CAR, we constructed GPC3-41BB-Z CAR in retroviral vector and generated GPC3-targeted CAR T cells. Th9-polarizing condition also significantly increased GPC3 CAR expression in T cells (new Fig. S10a). Flow cytometry showed that GPC3 Th9 and Tc9 CAR-T cells exhibited significantly enhanced IL9 (new Fig. S10b) and reduced IFN γ (new Fig. S10c) expressions. GPC3 T9 CAR T cells expressed high level of CCR7, indicating higher percentage of Tcm (CCR7⁺CD45RO⁺) cells compared to GPC3 T1 CAR T cells (new Fig. S10d). Similar to CD19 T9 CAR-T cells, GPC3 T9 CAR-T cells also showed a superior antitumor activity

compared with GPC3 T1 CAR-T cells (new Fig. 7i-j). Moreover, GPC3 T9 CAR-T cells persisted longer in peripheral blood and exhibited better tumor-infiltrating ability than GPC3 T1 CAR-T cells (new Fig. 7k-n). Thus, the new findings support our statement that Th9-polarized CAR-T cells are better effector T cells than classically IL2-activated CAR-T cells for cancer immunotherapy.

Major specific comments:

Were the Tc9 / Th9 grown separately or in bulk? and control Tc1/Th1 T-cells? If so – how were they separated? What was the transduction efficacy of the Tc9/Th9?

Response: They were grown in bulk and sorted by flow cytometry in order to perform RNA-seq experiment. The cells in each flow cytometry were pre-gated for GFP⁺CD3⁺CD8⁺ or GFP⁺CD3⁺CD4⁺ population. In our proliferation assay (Fig. 2a), cell size (Fig. S3), and new experiment (new Fig. S7), CD4⁺ or CD8⁺ T cells were isolated and cultured separately.

The transduction efficacy of Th9 and Tc9 cells are shown in new Fig. S1.

Specifically - in figure 1c-d, flow cytometry shows that only a subset of cells expresses IL9. Can you confirm these were the CAR-positive cells (by flow for example)? This is even more important given the in-vivo data (fig 5a-b, line 176-178), showing lack of IL9 expression in the cells in vivo. An alternative explanation would be that the IL9+ cells die in vivo and the 'regular' CAR+ T-cells persist.

Response: CAR expression in CAR-T cells was confirmed by gating GFP⁺CD3⁺CD4⁺ or GFP⁺CD3⁺CD8⁺ cells. We conducted a new experiment to examine the effect of IL9 on the efficacy of T9 CAR-T cells. We treated NAML6-bearing mice with IL9-neutralizing antibodies and injected with T9 CAR-T cells (new Fig. S8). Results showed that anti-IL9 did not significantly affect tumor growth or survival of tumor-bearing mice receiving T9 CAR-T cell treatment. Moreover, IL9 antibody did not significantly affect the persistence of T9 CAR T cells in vivo. The results demonstrate that IL9 is not a major player for antitumor efficacy of T9 CAR T cells, which is consistent with our previous study that OT-II Th9 cells did not require IL9 to exert their antitumor activity in mice (Lu et al, Cancer Cell 33:1048-1060, 2018). Overall, we used IL9 as a differentiation marker for in vitro preparation of T9 CAR-T cells.

Minor comments:

In the introduction - the overall survival in ALL treated with CAR T cells is not 1 year as stated, perhaps the median OS (line 43-45); also, the 30% EFS/OS long-term is with a short-durability CD28-based CAR. Data is lacking regarding long term results of 41BB-costimulated CAR T cells, known to have longer persistence. For DLBCL, seems that long-term outcome is ~40% EFS (Locke, Lancet Oncol 2018).

Response: Sorry for the oversight. We have corrected it.

In figure 1, where is the gate in the IFN γ flow-plot drawn? Can dot plots also be shown for IFN γ in 1d (such as in 1c for IL9)?

Response: Thank you for the suggestion. We changed it into dot plots and show it in new Fig. 1d.

The regular CAR T (T1CAR) cells seem to have increased EM phenotype (fig 3) despite having a 41BB co-stimulatory domain (these percentages seem to be appropriate to CD28-based CARs), unlike what is seen from different groups.

Response: Inclusion of 4-1BB in the CAR architecture promotes the outgrowth of CD8⁺ central memory T cells, whereas CAR-T cells with CD28 domains yield effector memory cells (Immunity 44:380-90, 2016). We examined CM and EM phenotype of CAR-T cells at both days 7 and 16, and we only showed data at day 16 in Fig. 3c-d in which T1 CAR-T had increased EM and decreased CM. However, at day 7 (new Fig. 3c-d), T1 CAR-T cells showed lower or comparable EM than CM cells. Moreover, new Fig. 3d shows that the percentage of CM cells decreased from 60% (day 7) to 18% (day 16) in Th1 CAR-T and from 45% (day 7) to 15% (day 16) in Tc1 CAR-T cells, which are similar to the results by Kawalekar et al (Immunity 44:380-90, 2016).

Increase in mitochondrial mass with T9 CAR is important. However, would be cautious stating this is convincing of a less-differentiated state (line 202) but could be In-vivo model.

Response: We appreciate the suggestion. We have deleted it.

-can you comment on the cause of relapse in these mice? Loss of persistence?

Response: We believe that tumor relapse could be caused by CAR-T inability to eliminate all cancer cells in vivo. T cell limited persistence or infiltration to tumor bed, immune suppressive tumor microenvironment, and/or tumor downregulated expression of targeted antigen could be the reasons.

Typos-

Many typos throughout of NALM6 (correct) vs NAML6 – please revise. Would recommend changing the beginning of the discussion, as age of the patient/donor is not a factor, and, for example, age of lymphocyte donors to this study was at all not mentioned. Please revise reference style as some are written differently (abstract format?) and some references are written with little justification (e.g. ref#3 Kantar Jian et al which has no relation to CAR therapy mentioned in this sentence).

Response: Sorry for the oversight. Based on the comments, we have carefully edited the manuscript to correct the mistakes.

REVIEWERS' COMMENTS

Reviewer #1 (Remarks to the Author):

The authors address all my concerns

Reviewer #2 (Remarks to the Author):

Authors have addressed my concerns adequately.

Reviewer #3 (Remarks to the Author):

The authors have answered all my questions in detail, and have significantly improved the manuscript to support their claims.

Only technical comment – please review figure s10 b and c – somehow Tc1 cells express IL9, likely a typo in headers.

Reviewers' Comments:

Reviewer #3:

Only technical comment – please review figure s10 b and c – somehow Tc1 cells express IL9, likely a typo in headers.

Response: Thanks for pointing out this typo, we have corrected it.